# Coupling chemosensory array formation and localization

Alejandra Alvarado[1], Andreas Kjær[1], Wen Yang[2], Petra Mann[1], Ariane Briegel[2], Matthew K Waldor[3,4,5], Simon Ringgaard[1]*

[1]Department of Ecophysiology, Max Planck Institute for Terrestrial Microbiology, Marburg, Germany; [2]Institute of Biology, Leiden University, Leiden, Netherlands; [3]Division of Infectious Diseases, Brigham and Women's Hospital, Boston, United States; [4]Howard Hughes Medical Institute, Harvard Medical School, Boston, United States; [5]Department of Microbiology and Immunobiology, Harvard Medical School, Boston, United States

**Abstract** Chemotaxis proteins organize into large, highly ordered, chemotactic signaling arrays, which in Vibrio species are found at the cell pole. Proper localization of signaling arrays is mediated by ParP, which tethers arrays to a cell pole anchor, ParC. Here we show that ParP's C-terminus integrates into the core-unit of signaling arrays through interactions with MCP-proteins and CheA. Its intercalation within core-units stimulates array formation, whereas its N-terminal interaction domain enables polar recruitment of arrays and facilitates its own polar localization. Linkage of these domains within ParP couples array formation and localization and results in controlled array positioning at the cell pole. Notably, ParP's integration into arrays modifies its own and ParC's subcellular localization dynamics, promoting their polar retention. ParP serves as a critical nexus that regulates the localization dynamics of its network constituents and drives the localized assembly and stability of the chemotactic machinery, resulting in proper cell pole development.
DOI: https://doi.org/10.7554/eLife.31058.001

**\*For correspondence:**
simon.ringgaard@mpi-marburg.mpg.de

**Competing interests:** The authors declare that no competing interests exist.

## Introduction

Chemotaxis is one of the primary means by which motile bacteria sense, respond, and adapt to changing environmental conditions. This process enables motile bacteria to perceive changes in local concentrations of chemicals; as a result, they can bias their movement away from unfavorable chemical stimuli and towards more favorable compounds (*Wadhams and Armitage, 2004*; *Sourjik and Armitage, 2010*). In the best studied model organism *Escherichia coli*, chemotaxis is mediated by an array of highly organized macromolecular complexes built from core chemotaxis units. The core units are themselves composed of a highly organized set of chemotaxis signaling proteins (*Figure 1A–B*). In general, the chemotaxis signaling cascade is initiated upon the detection of chemotactic stimuli by methyl-accepting chemotaxis proteins (MCPs). These membrane-spanning receptors then interact with a cytoplasmic histidine kinase, CheA, while the adaptor protein CheW stabilizes this interaction and participates in regulating CheA kinase activity (*Ortega et al., 2013*; *Parkinson et al., 2015*). A phosphosignaling cascade is initiated via CheA and its cognate response regulator CheY. Phosphorylated CheY induces a change in flagellar rotation and consequently in the direction of bacterial swimming, which over time results in net movement towards a more favorable environment (*Wadhams and Armitage, 2004*; *Sourjik and Armitage, 2010*).

MCPs usually consist of a variable N-terminal extracellular ligand binding domain, a cytoplasmic HAMP domain, and a well conserved signaling domain (or kinase control domain) with a highly conserved protein interaction tip that directs the assembly and action of receptor signaling complexes (*Figure 1A*) (*Kim et al., 1999*; *Falke and Hazelbauer, 2001*; *Alexander and Zhulin, 2007*;

**eLife digest** Many bacteria live in a liquid environment and explore their surroundings by swimming. When in search of food, bacteria are able to swim toward the highest concentration of food molecules in the environment by a process called chemotaxis. Proteins important for chemotaxis group together in large networks called chemotaxis arrays. In the bacterium *Vibrio cholerae* chemotaxis arrays are placed at opposite ends (at the "cell poles") of the bacterium by a protein called ParP. This makes sure that when the bacterium divides, each new cell receives a chemotaxis array and can immediately search for food. In cells that lack ParP, the chemotaxis arrays are no longer placed correctly at the cell poles and the bacteria search for food much less effectively.

To understand how ParP is able to direct chemotaxis arrays to the cell poles in *V. cholerae* Alvarado et al. searched for partner proteins that could help ParP position the arrays. The search revealed that ParP interacts with other proteins in the chemotaxis arrays. This enables ParP to integrate into the arrays and stimulate new arrays to form. Alvarado et al. also discovered that ParP consists of two separate parts that have different roles. One part directs ParP to the cell pole while the other part integrates ParP into the arrays. By performing both of these roles, ParP links the positioning of the arrays at the cell pole to their formation at this site.

The findings presented by Alvarado et al. open many further questions. For instance, it is not understood how ParP affects how other chemotaxis proteins within the arrays interact with each other. As well as enabling many species of bacteria to spread through their environment, chemotaxis is also important for the disease-causing properties of many human pathogens – like *V. cholerae*. As a result, learning how chemotaxis is regulated could potentially identify new ways to stop the spread of infectious bacteria and prevent human infections.

DOI: https://doi.org/10.7554/eLife.31058.002

*Hazelbauer et al., 2008*). Importantly, the tip contains sites for forming trimers of receptor dimers (*Kim et al., 1999*; *Parkinson et al., 2015*), and for binding to CheA and CheW (*Miller et al., 2006*; *Park et al., 2006*; *Vu et al., 2012*; *Wang et al., 2012*; *Li et al., 2013*; *Piasta et al., 2013*; *Pedetta et al., 2014*; *Cassidy et al., 2015*) (*Figure 1B*). The histidine kinase CheA is comprised of five separate domains (P1 to P5) with specific functions (*Figure 1B*, green). P1 is the phosphotransfer domain and contains the substrate histidine for autophosphorylation; P2 binds CheY for phospho-transfer from P1 (*Swanson et al., 1993*; *Morrison and Parkinson, 1994*; *Bilwes et al., 1999*); P3 is the dimerization domain (*Park et al., 2006*; *Cassidy et al., 2015*); P4 is the kinase or ATP binding domain; and P5 is an SH3-like regulatory domain, which binds the signaling tip of MCPs (*Borkovich et al., 1989*; *Gegner et al., 1992*; *Bilwes et al., 1999*; *Zhao and Parkinson, 2006*). The adaptor protein CheW (*Figure 1B*, red) consists of a single SH3-like domain, and is structurally simi-lar to P5 of CheA (*Griswold et al., 2002*; *Li et al., 2013*; *Cassidy et al., 2015*).

Together, MCPs, CheA, and CheW form stable core signaling complexes. As shown in *Figure 1B*, one CheA dimer joins two MCP trimer-of-dimers and two CheW proteins. The helix formed by the dimerization of the P3 domains of CheA positions itself between the two MCP dimer-of-trimers (*Briegel et al., 2011*; *Li and Hazelbauer, 2011*; *Briegel et al., 2012*; *Liu et al., 2012, 2013*; *Briegel et al., 2014a*) and each P5 domain of a CheA dimer binds to one CheW. Therefore, a single core unit is arranged in a hexagonal structure held together by contacts between: (i) CheA-MCPs, (ii) CheW-MCPs and (iii) CheA-CheW. According to the current model, further hexagonal core units then join to form a super-lattice structure, commonly known as the chemosensory array (*Briegel et al., 2009, 2012*; *Liu et al., 2012, 2013*; *Briegel et al., 2014a, 2014b*; *Piasta and Falke, 2014*) (*Figure 1B*). In vivo and in vitro observations indicate that CheA-CheW interactions bridge the two receptor trimers of every core and give the array its characteristic stability and high sensitivity (*Zhao and Parkinson, 2006*; *Hazelbauer et al., 2008*; *Erbse and Falke, 2009*; *Briegel et al., 2009*; *Li and Hazelbauer, 2011*; *Briegel et al., 2012*; *Slivka and Falke, 2012*; *Sourjik and Wingreen, 2012*; *Liu et al., 2012*; *Briegel et al., 2014a*; *Piasta and Falke, 2014*). However, while there is much knowledge of array structure, the mechanisms that underlie the formation and localization of these elaborate structures are incompletely understood, especially in systems other than *E. coli*.

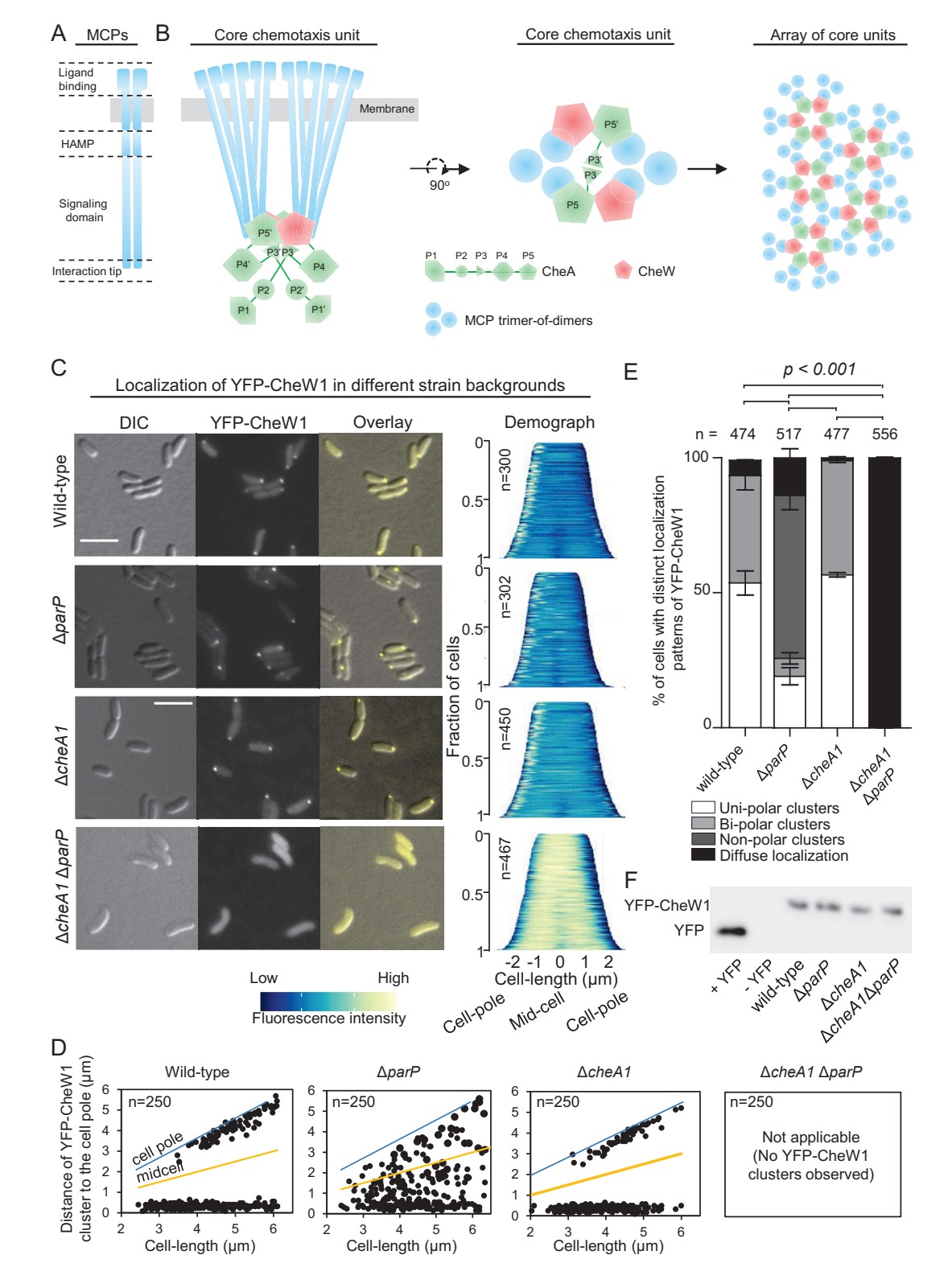

**Figure 1.** ParP contributes to signaling array formation. (**A**) Schematic of the domain architecture of the MCP dimer. (**B**) Schematic showing the structure of chemotaxis core units and how these units assemble into signaling arrays. (**C**) Fluorescence microscopy images showing the intracellular localization of YFP-CheW1 in wild-type and indicated *V. cholerae* mutant backgrounds. Demographs show the fluorescence intensity of YFP-CheW1 along the cell length in a population of *V. cholerae* cells relative to cell length. Scale bars represents 5 µm. (**D**) Graphs depicting the distance of YFP-
*Figure 1 continued on next page*

*Figure 1 continued*

CheW1 foci from the cell pole as a function of cell length. (**E**) Bar graph showing the percentage of cells with distinct YFP-CheW1 localization patterns in the indicated *V. cholerae* strain backgrounds. Error bars indicate standard error of the mean (SEM). The n-value indicates the total number of cells analyzed from three independent experiments. (**F**) Immunoblot using JL8 anti-YFP antibodies to detect the presence of YFP and YFP-CheW1 in *V. cholerae* strains imaged in (**C**). As a positive control, a strain expressing YFP from plasmid pMF390 was included (+YFP). A strain not expressing YFP (-YFP) was included as a negative control.

DOI: https://doi.org/10.7554/eLife.31058.003

The following figure supplement is available for figure 1:

**Figure supplement 1.** Chemotaxis arrays form in the absence of CheA.

DOI: https://doi.org/10.7554/eLife.31058.004

As mentioned above, chemotaxis has been extensively studied in *E. coli,* a peritrichously flagellated bacterium. Here, array formation is thought to be a stochastic process in which individual receptors are inserted randomly in the membrane, and subsequently diffuse freely until they either join existing arrays or nucleate new ones (*Thiem and Sourjik, 2008*). This process results in a non-uniform distribution of signaling arrays at cell poles and randomly along the cell length (*Sourjik and Berg, 2000*), and likely ensures that sensory arrays are in close proximity to the lateral flagella. In organisms such as *Caulobacter crescentus, Pseudomonas aeruginosa,* and *Rhodobacter sphaeroides,* and several *Vibrio* species, chemosensory arrays are actively localized to the cell poles (*Alley et al., 1992*; *Maddock and Shapiro, 1993*; *Wadhams et al., 2003*; *Bardy and Maddock, 2005*; *Ringgaard et al., 2011*; *Ringgaard et al., 2014*). In the polarly flagellated pathogens *Vibrio cholerae* and *Vibrio parahaemolyticus,* we recently reported that chemosensory arrays are exclusively localized at one or both cell poles by a mechanism that depends on the partner proteins ParC and ParP, both of which are encoded within the chemotaxis operon (*Ringgaard et al., 2011*; *Yamaichi et al., 2012*; *Ringgaard et al., 2014*). For *V. cholerae,* chemotaxis proteins encoded by chemotaxis operon II, e.g. CheA1 and CheW1, are directed to the cell pole by ParC and ParP (*Ringgaard et al., 2015*; *Briegel et al., 2016*), and from here on, CheA and CheW will be used instead of CheA1 and CheW1, respectively. In newborn *Vibrio* cells, these signaling arrays are exclusively localized to the old flagellated cell pole, then recruited to the new cell pole as cells enlarge, resulting in a bi-polar localization pattern. Thus, at cell division each daughter cell inherits a signaling array positioned at its old pole (*Ringgaard et al., 2011*, *2014*). In the absence of either ParC or ParP, the chemotaxis arrays are no longer properly recruited to the cell poles. Instead, signaling arrays form and localize randomly along the cell length, and bi-polar localization is not established prior to cell division. Therefore, daughter cells do not faithfully inherit a signaling array at their old poles, resulting in altered motility and decreased chemotaxis (*Ringgaard et al., 2011*, *2014*).

ParC mediates polar localization of ParP, which in turn interacts with a specific domain of CheA that is only present in CheA proteins with an associated ParC/ParP-system (CheA-LID) (*Ringgaard et al., 2014*). ParP prevents dissociation of CheA from chemotaxis arrays and disruption of either ParP-ParC or ParP-CheA interactions results in defective recruitment of chemotaxis arrays to the cell poles, leading to their random instead of polar localization (*Ringgaard et al., 2011*, *2014*). However, the molecular mechanisms by which this protein interaction network governs the dynamic localization of chemotactic signaling arrays remain to be elucidated. Notably, there is little knowledge of how factors promoting array positioning are able to access and guide localization of chemotaxis proteins. In particular, it is not clear how such factors are integrated within the widely conserved structure of signaling arrays.

Here, using *V. cholerae* as a model organism, we analyze how ParP is able to gain access to and interact with chemotaxis proteins positioned within the highly ordered structure of signaling arrays and how it mediates their intracellular localization. We identify MCP proteins as a new interaction partner for ParP. Via interactions with MCPs and CheA, ParP is a part of the chemotaxis core unit and integrates into the chemotactic signaling arrays. Importantly, ParP integrates into arrays and promotes their formation via a C-terminal Array Integration and Formation (AIF) domain, which is linked to ParP's N-terminal ParC interaction domain. Linkage of these domains within ParP couples array formation and localization and results in localized formation of arrays at the cell poles and thus promotes cell pole maturation.

## Results

### ParP contributes to signaling array formation

To address how ParP is able to access chemotaxis proteins within signaling arrays in *V. cholerae*, we analyzed array localization in wild-type, *cheA1, parP* and *cheA1 parP* deletion backgrounds using a functional (*Ringgaard et al., 2011*) YFP-CheW1 fusion as a marker for array localization and formation. In wild-type cells YFP-CheW1 mainly localized in clusters at the cell poles (*Figure 1C–E*). In contrast to localization in wild-type cells, in the absence of ParP, YFP-CheW1 clusters were not recruited to the cell poles, but were instead mislocalized along the cell length or completely absent in 74% of cells (*Figure 1C–E*). In a strain lacking *cheA1,* YFP-CheW1 still formed clusters at the cells poles in a manner indistinguishable to that observed in wild-type cells (*Figure 1C–E*), suggesting that chemotaxis arrays still form in the absence of CheA.

To analyze if arrays are still properly formed in the absence of CheA, we performed cryo-electron microscopy (cryo-EM) on wild-type and Δ*cheA* cells (*Figure 1—figure supplement 1*). For both strains, chemotaxis arrays were detectable and indistinguishable in structure, consisting of an inner membrane-anchored array of MCP proteins and an associated cytosolic baseplate. Out of 61 cells imaged with cryo-EM for each strain, there was a 60% reduction in the number of cells with observable arrays in the Δ*cheA* background compared to wild-type – consistent with a role of CheA in stimulating array formation. However, the cryo-EM experiments reveal that ordered signaling arrays can still form in the absence of CheA. Furthermore, these cryo-EM images strongly suggest that the YFP-CheW1 clusters reflect the localization and formation of properly structured arrays in the absence of CheA, although we cannot formally exclude the possibility that YFP-CheW1 clusters may reflect misformed or variant states of supramolecular complexes in some cells. Strikingly, in the double deletion strain Δ*cheA1 ΔparP*, YFP-CheW1 did not form clusters but was localized diffusely in the cytoplasm (*Figure 1CE*, bottom). Immunoblot analysis showed that the difference in localization of YFP-CheW1 was not due to differences in expression levels or cleavage of the YFP moiety from the YFP-CheW1 fusion construct (*Figure 1F*). These data indicate that formation of signaling arrays is severely compromised in the absence of both ParP and CheA, and that CheW1 alone only has a minor effect on array formation but requires the presence of either ParP or CheA, which individually are sufficient for promoting array formation. These data are supported by cryo-EM analyses of the Δ*cheA1 ΔparP* strain, in which out of 61 imaged cells there was an 85% reduction in the number of cells with detectable signaling arrays compared to wild-type. Together, these observations suggest that ParP participates in the process of array formation in addition to its previously known function in promoting polar localization of signaling arrays.

### ParP interacts with the signaling domain of methyl-accepting-chemotaxis proteins

To further investigate how ParP contributes to array formation and localization, we performed a screen to identify additional ParP interaction partners. We developed a bacterial-two-hybrid blue/white-colony screen in *E. coli*, using ParP as bait against a chromosomal library from *V. cholerae* (*Figure 2A* and *Figure 2—figure supplement 1*). Bacteria harboring a plasmid expressing a ParP interaction partner give rise to blue colonies (*Figure 2A*). It is important to note that *E. coli* does not encode homologs of either ParP or ParC, thus reducing the possibility of indirect interactions mediated by an endogenous *E.coli* factor, and suggesting that interaction partners identified in this assay likely interact directly with ParP. One hundred blue colonies were picked and the candidate ParP interaction partners identified by sequencing. Of the 100 blue colonies sequenced, 95 contained plasmids with genes encoding MCP proteins, corresponding to 15 distinct MCPs (*Figure 2B*). While the fragments of all the *mcp* genes hit in the screen covered varying regions of the respective genes, all hits included the regions encoding the signaling domains of the MCP proteins. Therefore, we assessed whether signaling domains (including the conserved interaction tip) from four MCPs were sufficient to mediate interactions with ParP (*Figure 2C*). All four MCP signaling domains interacted with ParP (*Figure 2C*), confirming that MCPs are a newly identified ParP interaction partner and that interaction occurs via the MCP signaling domain. No interaction between ParC and MCPs was observed, suggesting only ParP, but not ParC interacts with MCP proteins.

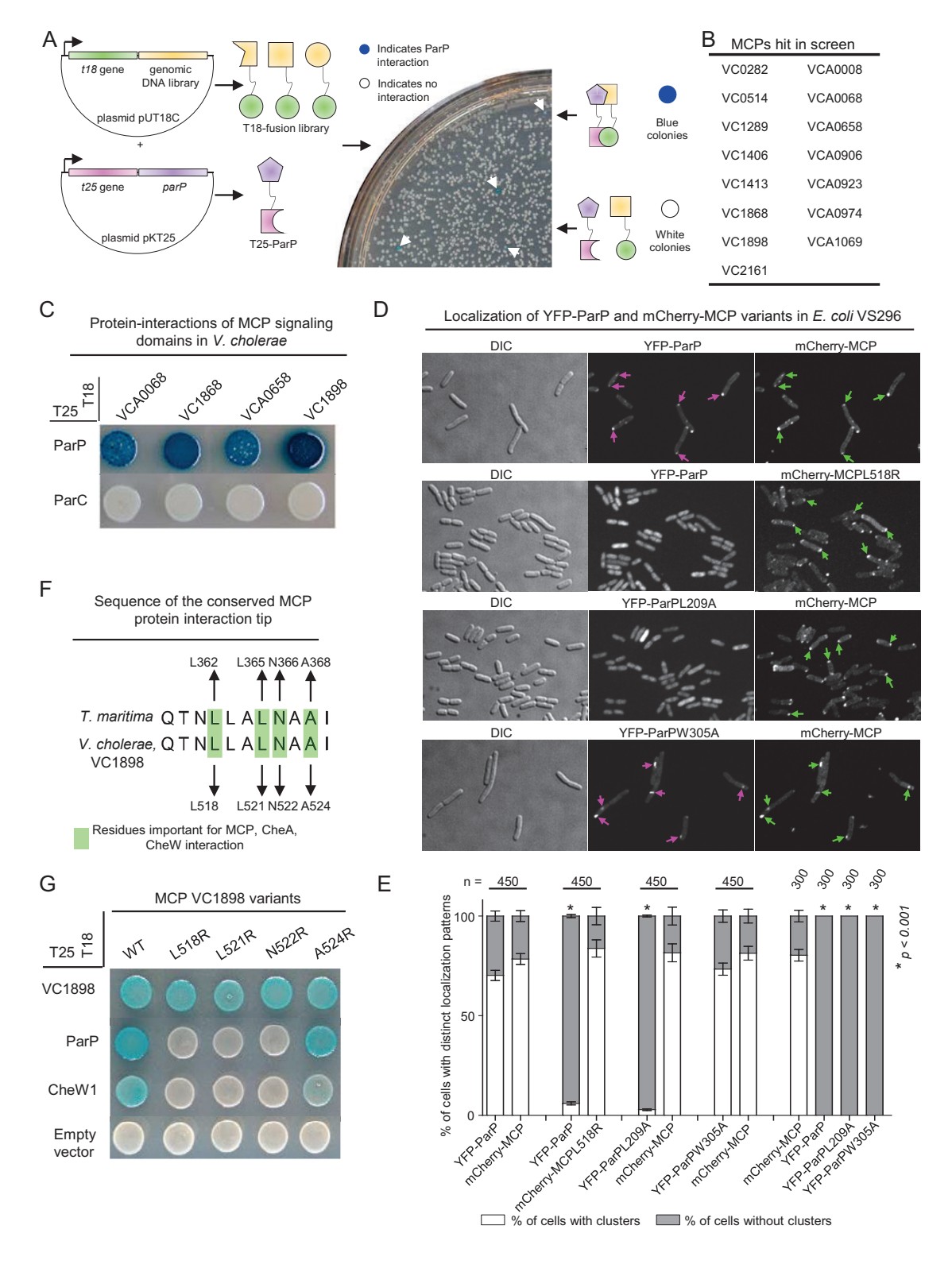

**Figure 2.** ParP interacts with the protein interaction tip of methyl-accepting-chemotaxis proteins. (**A**) To screen for ParP interaction factors, *E. coli* strain BTH101 carrying plasmid pAK08 (encoding T25-ParP) was transformed with the plasmid library and spread on indicator plates. Bacterial colonies encoding a candidate ParP interaction partner turned blue. Blue colonies were picked and the chromosomal DNA inserted in the pUT18 vector was identified by sequencing. (**B**) Summary of the MCP proteins identified as ParP interaction partners in the bacterial-two-hybrid (BACTH) screen. (**C**)
*Figure 2 continued on next page*

*Figure 2 continued*

BACTH experiment assaying for protein interactions of the *V. cholerae* MCP signaling domains (SD) of MCPs VCA0068, VC1868, VCA0658, and VC1898 with ParP and ParC. Blue coloration of bacterial colonies indicates an interaction. (**D**) Fluorescence microscopy of YFP-ParP and mCherry-MCP VC1898 (mCherry-MCP) variants in *E. coli* strain VS296. Purple arrows indicate clusters of YFP-ParP variants. Green arrows indicate clusters of mCherry-MCP. (**E**) Bar graphs indicate the percentage of cells with clusters of YFP-ParP variants and mCherry-MCP in *E. coli* VS296. Error bars indicate standard error of the mean (SEM). The n-value indicates the total number of cells analyzed from three independent experiments. Asterisks indicate p<0.001 compared to VS296 co-expressing wild-type YFP-ParP with mCherry-MCP. (**F**) Alignment of the conserved protein interaction tips of MCP TM1143 from *T. maritima* and MCP VC1898 of *V. cholerae*. Highlighted *V. cholerae* amino acids were chosen as candidates for amino acid substitution. (**G**) BACTH experiment assaying for protein interactions of *V. cholerae* MCP VC1898 and its variants.

DOI: https://doi.org/10.7554/eLife.31058.005

The following figure supplements are available for figure 2:

**Figure supplement 1.** Bacterial-two-hybrid screen for identification of ParP interaction partners.

DOI: https://doi.org/10.7554/eLife.31058.006

**Figure supplement 2.** YFP-ParP is diffusely localized to the cytoplasm in *E. coli*.

DOI: https://doi.org/10.7554/eLife.31058.007

**Figure supplement 3.** Alignment of the MCP protein interaction tip of MCPs from *V. cholerae*.

DOI: https://doi.org/10.7554/eLife.31058.008

To test whether ParP and MCP could interact independently of other chemotaxis proteins, we co-expressed YFP-ParP and mCherry-MCP-VC1898 (denoted mCherry-MCP) and assayed for co-localization in an *E. coli* strain deleted for all native chemotaxis proteins (strain VS296). When expressed alone, YFP-ParP was diffusely localized in the cytoplasm in 100% of cells, and mCherry-MCP localized as distinct clusters (*Figure 2—figure supplement 2*). Strikingly, when YFP-ParP was co-expressed with mCherry-MCP, YFP-ParP also localized in clusters that always co-localized with mCherry-MCP clusters (*Figure 2D–E*). Therefore, in addition to interacting with CheA and ParC, ParP also interacts (likely in a direct fashion) with MCP proteins. Since ParP interacts with both CheA and MCPs, we hypothesize that ParP forms part of the core chemotaxis unit.

## The MCP protein interaction tip mediates interaction with ParP

Next, we investigated which MCP residues are required for MCP-ParP interaction. Interestingly, the C-terminal part of ParP consists of a predicted SH3-like domain (hereafter named array integration and formation domain – AIF domain) similar to CheW and the P5 domain of CheA. The highly conserved protein interaction tip within the MCP signaling domain is responsible for interactions with CheW and CheA-P5 proteins (*Kremer et al., 1996*; *Kim et al., 1999*; *Li et al., 2007*, *2011*; *Li and Hazelbauer, 2011*; *Briegel et al., 2012*; *Liu et al., 2012*, *2013*; *Cassidy et al., 2015*). Furthermore, several residues in the MCP TM1143 from *T. maritima* have been shown to be important for these interactions: L362, L365, N366, and A368 (*Figure 2F*). Multiple sequence alignment of all predicted *V. cholerae* MCPs with the sequence of MCP TM1143 from *T. maritima*, revealed that these four residues are conserved in all of the MCPs identified in the two-hybrid screen and all but two putative MCPs found in *V. cholerae* (*Figure 2—figure supplement 3*). We chose MCP VC1898, the MCP with the strongest signal for interaction with ParP, to create individual amino acid substitution variants (VC1898-L518R, L521R, N522R, A524R; *Figure 2F*) and tested their interaction capabilities. Three of the four substitutions (L518R, L521R, and N522R) disrupted the capacity of the MCP to interact with CheW1, but not with itself (*Figure 2G*). Notably, the same substitutions also abolished the interaction between the MCP and ParP (*Figure 2G*). Since the MCP variants retained the ability to self-interact, the effect on their interactions with CheW1 and ParP is likely not due to reduced expression levels of the MCP variants. Moreover, we tested the L518R variant for interaction with ParP in the *E. coli* VS296 co-expression assay. Notably, YFP-ParP no longer formed clusters co-localizing with mCherry-MCP-L518R clusters, but instead localized diffusely in the cytoplasm (*Figure 2D*) in 95% of cells (*Figure 2E*), indicating that the L518R substitution abrogates the capacity of YFP-ParP and mCherry-MCP to interact. Altogether, these observations suggest that ParP-AIF targets the same MCP residues that mediate MCPinteractions with CheW and CheA, and thus lends support to the idea that ParP is a component of the chemotaxis core unit of signaling arrays.

## A conserved hydrophobic pocket within the SH3-like domain of ParP mediates interaction with MCP signaling domains

While ParP-AIF domains form their own distinct clade of SH3-domains, they are more similar to the P5 domain of CheAs than to CheWs (*Figure 3A*). CheW and CheA-P5 are each composed of two subdomains (1 and 2) and the junction between the two subdomains contains branched hydrophobic residues that form a groove mediating interaction with the MCP interaction tip (in CheW from *Thermotoga maritima* MSB8: V27, I30, L14, V33; *Figure 3B*, red residues) (*Griswold et al., 2002*; *Park et al., 2006*; *Briegel et al., 2012*; *Li et al., 2013*). The AIF domain of ParP is predicted to have a similar overall protein architecture as CheA-P5 and CheW, and we hypothesized that the corresponding hydrophobic amino acids (L196, L209, L212A, and I215) in the junction between its putative subdomains function to promote ParP's interactions with MCPs (*Figure 3—figure supplement 1*). We replaced each of these amino acid residues with alanine, and evaluated each variant ParP's capacity to interact with the MCP signaling domain (MCP-SD; *Figure 3C*). L196A, L209A, and to some extent L212A (but not I215A), showed reduced interaction with the MCP-SDs, supporting that ParP interacts with the MCP protein interaction tip via residues in its putative interaction groove, in a manner similar to the way in which CheW and CheA-P5 interact with the MCPs. Notably, this hybrid assay suggested that replacement of L209 with alanine (ParPL209A) completely disrupted interaction between ParP and MCP-SD (*Figure 3C*). Additionally, in the *E. coli* co-expression assay, YFP-ParPL209A did not co-localize with mCherry-MCP clusters, but were instead localized diffusely in the cytoplasm (*Figure 2D–E*), further indicating that this residue is involved in mediating ParP-MCP interactions. Thus, ParP appears to rely on analogous residues as CheW and CheA-P5 to interact with MCPs. Interestingly, L196A, L209A, and L212A are almost 100% conserved amongst ParP proteins, suggesting it is a general property of ParP proteins to interact with the MCP-SD (*Figure 3—figure supplement 2*).

## Distinct ParP interfaces mediate its interaction with MCP and CheA

We next turned to analyzing the interaction between ParP and CheA. Previous work had revealed that a single amino acid in *V. parahaemolyticus* ParP was critical for interaction with CheA (*Ringgaard et al., 2014*), and we found that the corresponding amino acid in *V. cholerae* ParP (W305) lies within the AIF domain. *V. cholerae* ParPW305A did not interact with CheA; however, the single amino acid substitution in this variant had little influence on ParP's capacity to interact with the MCP in the two-hybrid assay (*Figure 3D*) or in the *E. coli* co-expression assay (*Figure 2D–E*). Conversely, although ParPL209A did not interact with MCPs, it was still capable of interacting with CheA (*Figure 3D*). ParP carrying both substitutions (ParPL209A-W305A, denoted ParP2PM) did not interact with either CheA or the MCP (*Figure 3D*). Neither substitution – either singly or in combination – impeded ParP's interactions with ParC (*Figure 3D*), which is mediated by an N-terminal domain that is separated from AIF by a long proline-rich linker (*Ringgaard et al., 2014*) (*Figure 3E*, *Figure 3—figure supplement 2*). Furthermore, since ParPL209A and ParPW305A were still observed to robustly interact with ParC, the effects observed on these variants' capacities to interact with MCP and CheA proteins (*Figure 3D*) are not likely explained by their decreased expression.

Based on the similarity of ParP to CheW of *Thermotoga maritima* MSB8, L209 and W305 are predicted to be positioned on opposite sides of the AIF domain (*Figure 3B*, *Figure 3—figure supplement 1*), supporting the idea that ParP-AIF contains distinct interfaces that direct its interactions with CheA-LID and MCPs, respectively (*Figure 3D*). Since ParP's N-terminus mediates interaction with ParC (*Ringgaard et al., 2014*), ParP has at least three distinct interaction interfaces. These distinct interaction surfaces potentially allow ParP to simultaneously couple two critical signaling components (MCP and CheA) to the polar determinant ParC (*Figure 3E*). Thus, ParP is a protein of high connectivity upon which both the chemotactic signaling network, as well as the system responsible for cell pole development, depend (*Figure 3E*).

## Interaction with MCPs or CheA is required for association of ParP with signaling arrays

We monitored localization of YFP-ParP and its variants co-expressed with CFP-CheW1 (a marker of arrays), to address whether ParP interactions with MCPs and/or CheA are required for its capacity to associate with signaling arrays. These experiments were done in a *V. cholerae ΔparC* background in

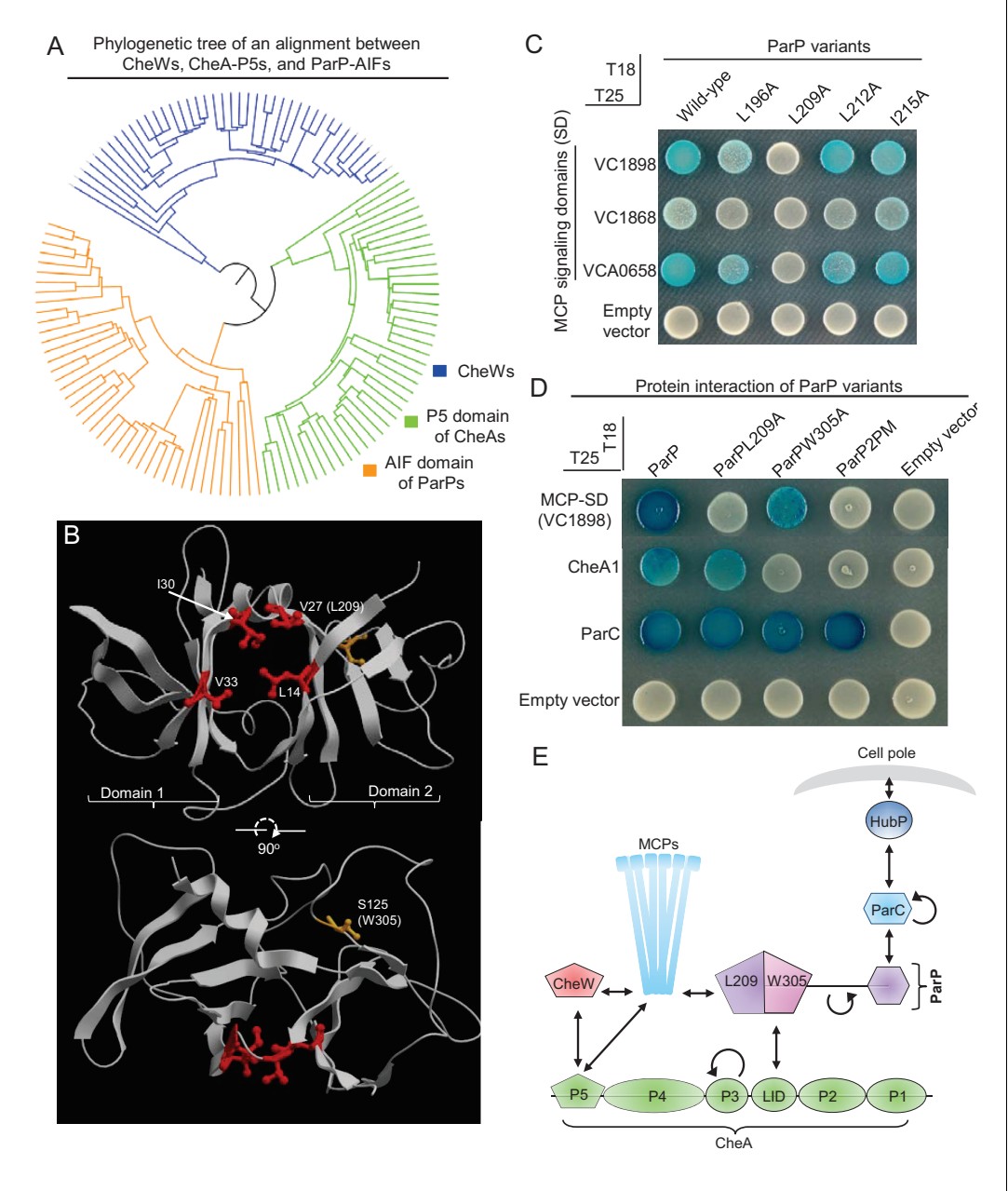

**Figure 3.** Two distinct interaction interfaces of ParP-AIF mediate its interaction with MCP and CheA respectively. (**A**) Phylogenetic tree of the SH3-like domains of CheW, CheA-P5 and ParP-AIF proteins. (**B**) Structure of *Thermotoga maritima* MSB8 CheW (PDB 3UR1, [*Briegel et al., 2012*]). CheW consists of two subdomains (1 and 2) responsible for interaction with subdomains 2 and 1 of the P5 domain of CheA. The junction between the two subdomains consists of branched hydrophobic residues (amino acids highlighted in red) that form a groove where CheW interacts with the MCP signaling domain helix. The corresponding amino acid L209 of ParP-AIF is noted in parentheses. The amino acid (S125) corresponding to the position of W305 in ParP-AIF is highlighted in orange. (**C**) BACTH experiment assaying interaction between ParP variants carrying amino acid substitutions in the predicted MCP binding pocket and MCP proteins VC1898, VC1868, and VCA0658. (**D**) BACTH experiment assaying interaction between ParP variants and MCP VC1898, CheA1, and ParC. (**E**) Schematic depicting ParP's three interaction interfaces, which enable the protein to interact with MCPs, CheA and ParC. L209 and W305 refer to amino acids important for interaction with MCP-SD and CheA respectively, within the two interaction interfaces of ParP-AIF.

DOI: https://doi.org/10.7554/eLife.31058.009

The following figure supplements are available for figure 3:

**Figure supplement 1.** ParP contains a C-terminal SH3-like domain.

DOI: https://doi.org/10.7554/eLife.31058.010

*Figure 3 continued on next page*

*Figure 3 continued*

**Figure supplement 2.** Residues responsible of MCP and CheA interaction are highly conserved amongst ParP proteins.

DOI: https://doi.org/10.7554/eLife.31058.011

order to investigate ParP's association with arrays without interference from its interactions with ParC. In ~75% of cells, YFP-ParP and CFP-CheW1 formed co-localized clusters (*Figure 4A,B*). Similarly, in ~50–55% of cells, CFP-CheW1 clusters were co-localized with those of YFP-ParPL209A or YFP-ParPW305A (*Figure 4A,B*). Thus, ParP's association with signaling arrays can be mediated by its interaction with either MCPs or CheA, though its capacity to interact with both these array components likely enhances its association with arrays. In striking contrast, when we expressed a ParP variant carrying both amino acid substitutions L209A and W305A (ParP2PM), which is unable to interact with either CheA or MCPs, fused to YFP (YFP-ParP2PM), almost no YFP-ParP2PM clusters were observed, despite the presence of CFP-CheW1 clusters in ~55% of cells. (*Figure 4A,B*). This result suggests that ParP's association with chemotaxis signaling arrays is fully dependent upon its interactions with CheA and MCPs. Furthermore, consistent with a function for ParP in stimulating array formation via its interactions with MCPs and CheA, there was a significant drop from ~75% of wild-type cells with YFP-CheW1 clusters, compared to ~50–55% only in cells expressing the ParPL209A, ParPW305A andParP2PM variants, respectively (*Figure 4A,B*).

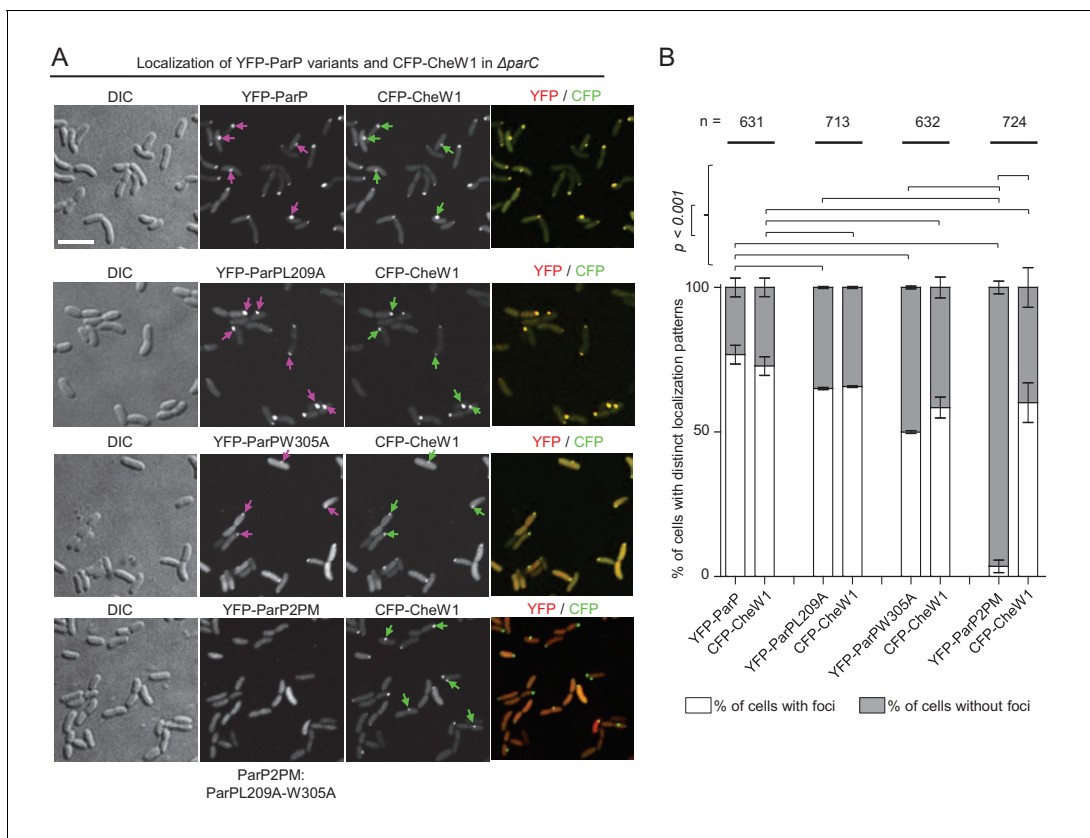

**Figure 4.** ParP's interaction with MCPs or CheA is required for its association with signaling arrays. (**A**) Fluorescence microscopy showing the intracellular localization of YFP-ParP variants and CFP-CheW1 in *V. cholerae* Δ*parC*. Scale bar represents 5 μm. Purple arrows indicate clusters of YFP-ParP variants. Green arrows indicate CFP-CheW1 clusters and pink arrows indicate YFP-ParP clusters. (**B**) Bar graphs indicate percentage of cells with foci of YFP-ParP variants and of CFP-CheW1 in *V. cholerae* Δ*parC*. Error bars indicate standard error of the mean (SEM). The n-value indicates the total number of cells analyzed from three independent experiments. (**A–B**) ParP2PM refers to a ParP variant carrying both the L209A and W305A amino acid substitutions.

DOI: https://doi.org/10.7554/eLife.31058.012

## The AIF domain of ParP is responsible for promoting signaling array formation

Our data indicate that either ParP or CheA are required for array formation. Consistent with the idea that the AIF domain accounts for ParP's activity in array formation, in the absence of CheA, chemotaxis clusters (as visually detected by YFP-CheW1) did not form in strains deleted for either the whole entire *parP* gene (Δ*parP* Δ*cheA1*) or only the ParP-AIF domain (*parP-ΔAIF* Δ*cheA1*) (**Figure 5A–C**). Moreover, the ParP variant with an AIF-domain incapable of integrating into signaling arrays (*parP2PM*) was almost entirely incapable of stimulating formation of chemotaxis clusters in the absence of CheA1 (strain *parP2PM* Δ*cheA1*) (**Figure 5A–C**).

In similar analyses, we investigated which CheA domain promotes its recruitment into signaling arrays and found that the P5 domain is both required and sufficient for recruitment of CheA into signaling arrays (**Figure 5D**). Absence of the CheA-P5 domain alone (*cheA1-ΔP5*) did not significantly influence array formation, however, combining deletion of CheA-P5 with deletion of ParP (*cheA1-ΔP5* Δ*parP*) also led to diffuse localization of YFP-CheW1 and consequently no formation of chemotaxis clusters (**Figure 5A–C**). This indicates that CheA stimulates arrays formation via its P5 domain, and further supports that the presence of ParP alone is sufficient for stimulation of array formation. Immunoblot analyses showed that the diffuse localization of YFP-CheW1 was not due to cleavage of the YFP moiety from the YFP-CheW1 fusion construct (**Figure 5—figure supplement 1**). Taken together, these data indicate that the AIF domain of ParP promotes formation of signaling arrays via its interactions with MCPs and CheA as an integral part of the core unit.

## ParP's N-terminal ParC interaction domain couples array localization to array formation

If ParP enables polar localization of chemotaxis clusters by integrating into the core chemotaxis unit, we reasoned that fusion of ParP's ParC-interaction domain to a different integral component of the core unit might also be capable of recruiting the chemotaxis clusters to the pole. To test this hypothesis, we constructed a ParP variant in which the AIF-domain was swapped for the CheA P5-domain in a Δ*cheA1* background (**Figure 5—figure supplement 2**, strain *parP-P5/ΔcheA1*), and tested for array localization by imaging YFP-CheW1 (**Figure 5B–C**). Indeed, the presence of ParP-P5 restored localization of uni- and bipolar clusters in 65% of cells, compared to 0% in a Δ*parP/ΔcheA1* background (**Figure 5B–C**). Thus, the ParC-interaction domain of ParP is capable of mediating polar localization of signaling arrays independent of the AIF-domain if fused to a protein that is part of the chemotaxis core unit and participates in array formation and structure (the CheA P5-domain). Collectively these observations suggest that ParP's capacity to localize arrays at the cell pole (mediated by its ParC-interaction domain) can operate independently of its capacity to promote array formation (mediated by AIF), and thus that ParP couples two distinct and separable functions.

## Integration of ParP within signaling arrays is required for their polar localization and inheritance

To test if the incorporation of ParP into signaling arrays and its facilitation of array formation had functional consequences on the polar localization of arrays, the localization of signaling arrays was determined in a set of ParP interaction mutants. Strain *parP2PM*, which produces the ParP2PM variant defective in interactions with both CheA and MCPs, exhibited a phenotype similar to that of Δ*parP*, with 65% of cells having mislocalized or absent arrays (**Figure 6A–B**). This deficiency in polar array localization, which is expected to preclude each daughter cell inheriting an array upon cell division, was largely dependent on ParP being unable to interact with both interaction partners; strains expressing a ParP variant defective in interaction solely with MCPs (ParPL209A) or CheA (ParPW305A) had a modest increase in mislocalized or absent arrays (9% and 20% of cells, respectively, compared to ~6% in wild-type cells) (**Figure 6A–B**). These data suggest that integration of ParP into signaling arrays either via interaction with MCPs or CheA, though compromised, to some extent can suffice to enable ParP-mediated array formation and polar localization. However, disruption of ParP's interaction with both MCPs and CheA, and thus its integration into the arrays, results in defective recruitment of arrays to the cell pole. Thus, ParP acts as an integral part of signaling arrays to couple the formation of signaling arrays and their polar localization, thereby ensuring their proper inheritance.

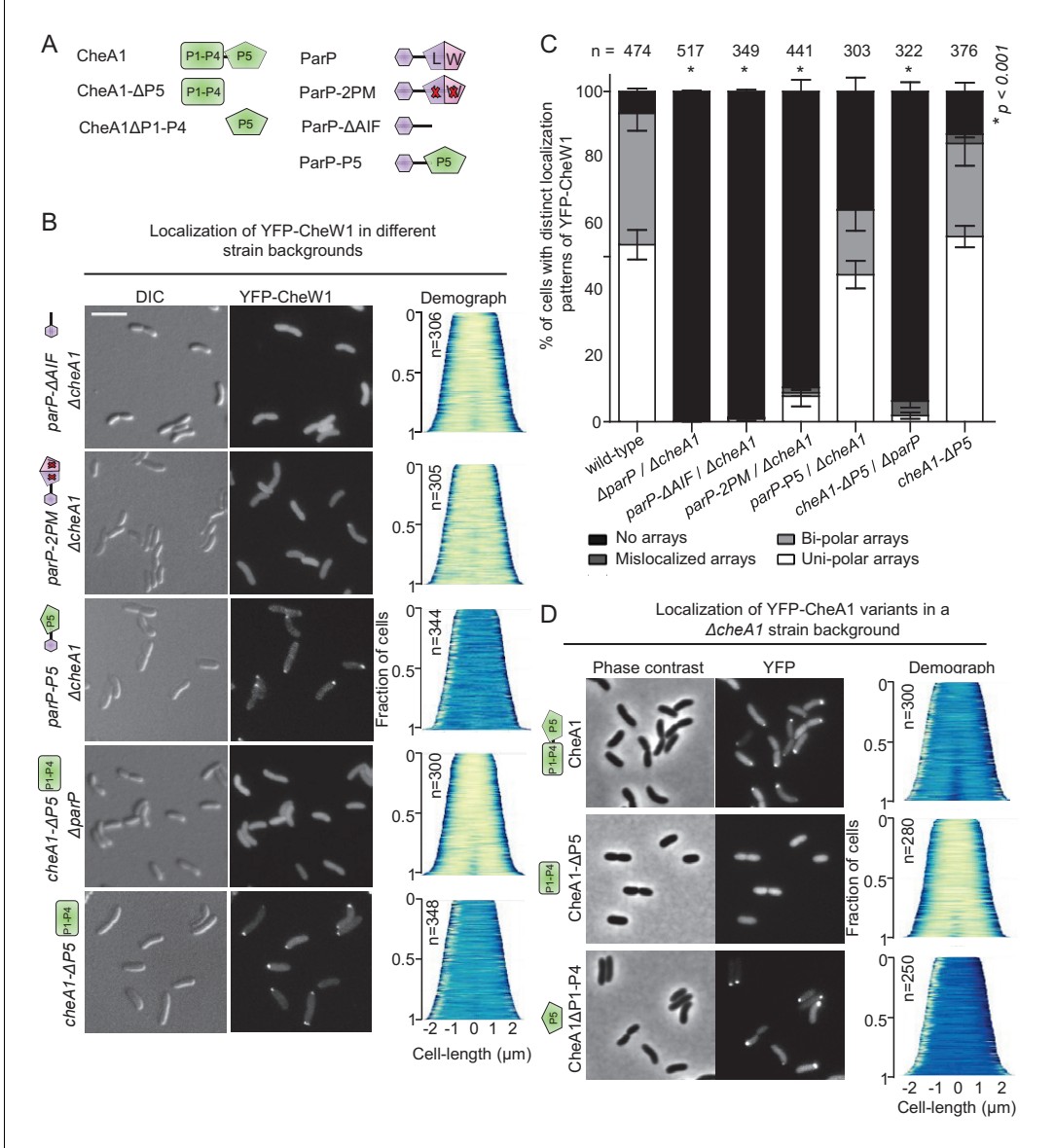

**Figure 5.** The ParP AIF domain and CheA-P5 promote formation of signaling arrays. (**A**) Schematic depicting the various CheA and ParP variants analyzed. (**B**) Fluorescence microscopy showing the intracellular localization of YFP-CheW1 in the indicated *V. cholerae* strain backgrounds. Scale bar represents 5 μm. (**C**) Bar graph showing the percentage of cells with distinct YFP-CheW1 localization patterns in the indicated *V. cholerae* strain backgrounds. Error bars indicate standard error of the mean (SEM). The n-value indicates the total number of cells analyzed from three independent experiments. Asterisks indicate p<0.001 compared to wild-type. (**D**) Fluorescence microscopy showing the intracellular localization of full-length and truncated versions of CheA1 fused to YFP in a Δ*cheA1* strain background. CheA: full-length CheA protein; CheA-(**P1–P4**): truncated version of CheA consisting of domain P1 to P4; CheA-P5: truncated version of CheA only consisting of the P5 domain.
DOI: https://doi.org/10.7554/eLife.31058.013

The following figure supplements are available for figure 5:

**Figure supplement 1.** YFP-CheW1 protein is stable in the analyzed *V. cholerae* strain backgrounds.
DOI: https://doi.org/10.7554/eLife.31058.014
**Figure supplement 2.** Domain swapping.
DOI: https://doi.org/10.7554/eLife.31058.015

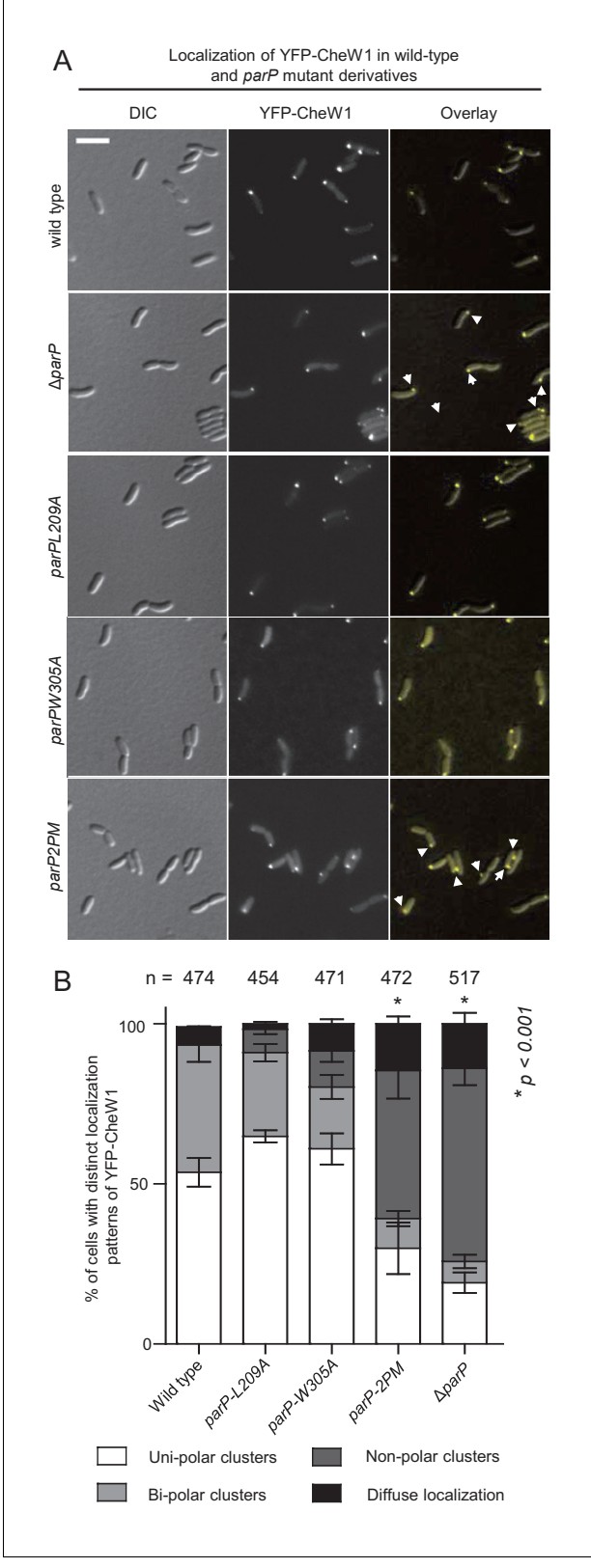

**Figure 6.** Interactions between ParP, MCPs, and CheA ensure proper polar localization and inheritance of signaling arrays. (**A**) Fluorescence microscopy images showing the intracellular localization of YFP-CheW1 in wild-type and different *V. cholerae parP* mutant backgrounds. Arrows indicate non-polar clusters of YFP-CheW1. Scale bar represents 5 μm. (**B**) Bar graph showing percentage of cells with distinct localization patterns of YFP-CheW1 in

*Figure 6 continued on next page*

*Figure 6 continued*
wild-type and different *V. cholerae* mutant backgrounds. Error bars indicate standard error of the mean (SEM). The n-value indicates the total number of cells analyzed from three independent experiments. Asterisks indicate p<0.001 compared wild-type.
DOI: https://doi.org/10.7554/eLife.31058.016

## Interactions between ParP, CheA and MCPs regulate polar localization of ParP

We next tested if ParP's interactions with MCP and CheA influenced the intracellular localization of ParP itself. Wild-type ParP and its variants ParPL209A, ParPW305A, and ParP2PM were fused to the C-terminus of YFP and expressed ectopically in a ΔparP strain background. Wild-type YFP-ParP localized to the cell poles in a uni- or bi-polar manner in 97% of cells. Consistent with their ability to still interact with ParC (*Figure 3D*), ParPL209A, ParPW305A, and ParP2PM localized as clusters at the cell pole in about 60% of all cells. However, in contrast to wild-type YFP-ParP, a significant proportion (~40%) of cells only showed diffuse localization of the YFP-ParP variants whereas wild-type ParP was diffuse in only 3% of cells (*Figure 7A–B*). Furthermore, a larger proportion of ParPL209A, ParPW305A, and ParP2PM were diffusely localized in the cytoplasm and there was a significant reduction in the intensity of these YFP-ParP variants at the cell pole compared to wild-type YFP-ParP (*Figure 7C*). Thus, interactions of ParP with both CheA and MCPs promote proper polar localization of ParP, and disruption of either interaction results in a decreased proportion of ParP being tethered to the cell pole – even when interactions to recruit chemotaxis arrays to this site appear sufficient to some extent (*Figure 6*).

## Integration of ParP within signaling arrays promotes its retention at the cell pole

To determine the underlying reason for reduced polar localization of ParP variants incapable of interaction with MCPs and CheA, we analyzed the recruitment and release of ParP and ParP2PM to and from the cell pole respectively. We performed FRAP (fluorescence-recovery-after-photobleaching) analysis on YFP-ParP and YFP-ParP2PM, to monitor the recruitment of new ParP molecules to the cell pole. After photobleaching of polar YFP-ParP and YFP-ParP2PM polar foci, we monitored the recovery of polar YFP fluorescence (*Figure 7D–E*). These experiments showed that there was a continuous recruitment of new ParP and ParP2PM from the cytoplasm to the cell pole, however, no significant difference in recovery rate was observed between the two ParP variants (*Figure 7D–E*). Next we measured the release of YFP-ParP and YFP-ParP2PM from polar clusters by bleaching the cytoplasmic signal from YFP-ParP and YFP-ParP2PM in cells with uni-polarly localized foci. The intensity of polar clusters was subsequently measured and plotted relative to the initial intensity as a function of time (*Figure 7F–G*). Post-bleach, the intensity of polar YFP-ParP and YFP-ParP2PM clusters decreased over time, demonstrating that both protein versions are continuously released from the polar clusters. However, the decay curves for the two ParP variants differed significantly: ParP reached a steady state after about 5 min, while ParP2PM was released at a faster rate than wild-type ParP, and the YFP-ParP2PM intensity continued to drop for over 11 min. This suggests ParP2PM is released from polar clusters to the cytoplasm to a much greater extent than wild-type ParP. Together, these experiments show that there is a continuous release of ParP molecules from the pole to the cytoplasm and recruitment of new ParP from the cytoplasm to the cell pole. Moreover, they reveal that ParP's capacity to interact with MCPs and CheA (and thereby integrate into signaling arrays) prevents its release, and as such promotes its retention, at the cell pole and consequently stabilizes its localization at this site.

## Integration of ParP within signaling arrays is required for proper polar localization of ParC

We also tested if ParP's ability to interact with the chemotaxis proteins MCP and CheA influenced the intracellular localization of the polar localization determinant ParC by ectopically expressing a functional YFP-ParC fusion protein (*Ringgaard et al., 2011*) in wild-type and *parP2PM* background strains. As previously reported, YFP-ParC localized in foci at the cell poles in wild-type *V. cholerae*

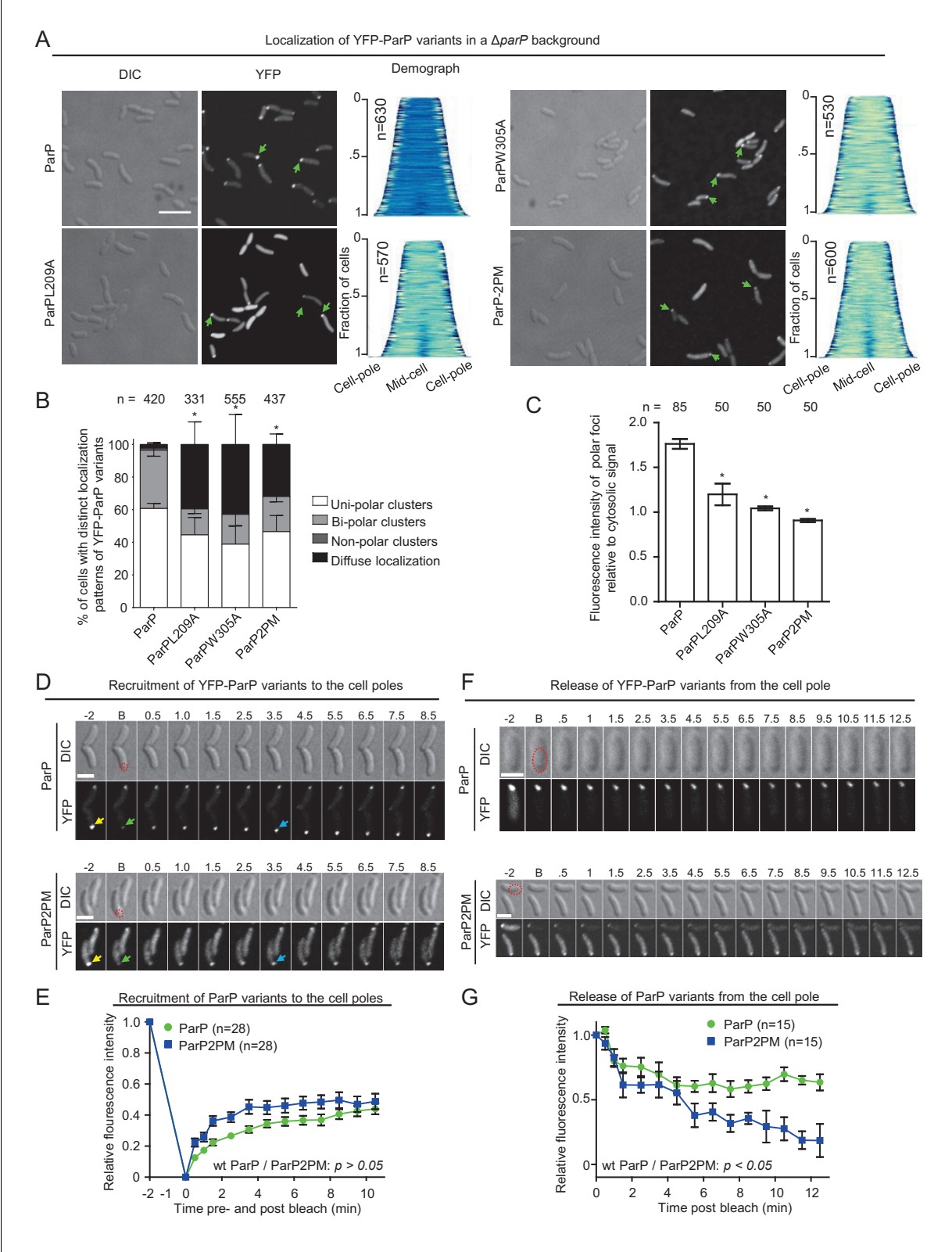

**Figure 7.** Integration of ParP within signaling arrays stabilizes recruitment of ParP to the cell pole. (**A**) Fluorescence microscopy images showing the intracellular localization of YFP-ParP variants in a *V. cholerae* Δ*parP* background. Demographs show the fluorescence intensity of YFP along the cell length in a population of *V. cholerae* cells relative to cell length. Scale bar represents 5 μm. Green arrows indicate polarly localized ParP clusters. (**B**) Bar graph showing the percentage of cells with distinct YFP-ParP localization patterns in the indicated *V. cholerae* strain backgrounds. (**C**) Bar graph

*Figure 7 continued on next page*

*Figure 7 continued*

showing the fluorescence intensity of polar ParP foci relative to cytosolic signal. (**B–C**) Error bars indicate standard error of the mean (SEM). The n-value indicates the total number of cells analyzed from three independent experiments. Asterisks indicate p<*0.001* compared to wild-type. (**D**) Fluorescence-recovery-after-photobleaching (FRAP) experiment of YFP-ParP and YFP-ParP2PM clusters at the cell poles showing that clusters recover post-bleaching. Numbers indicate minutes pre- and post-bleach. 'B' indicates bleaching. The red dashed circle shows the bleached region. Yellow arrows indicate the pre-bleach cluster, green arrows indicate the bleached cluster. Blue arrows indicate clusters with recovered YFP signal. (**E**) Graph depicting the fluorescence intensity of YFP-ParP and YFP-ParP2PM pre- and post-bleach at the bleached cell pole relative to the initial intensity at the pole pre-bleach during time-lapse series. The average recovery from 28 distinct cells is shown. Error-bars indicate standard error mean (SEM). (**F**) Release of YFP-ParP and YFP-ParP2PM from the cell pole post-bleach of the cytoplasmic signal. Numbers indicate minutes pre- and post-bleach. 'B' indicates bleaching. The red dashed circle shows the bleached region. (**G**) Graph depicting the fluorescence intensity of polar YFP-ParP and YFP-ParP2PM clusters post-bleach relative to the initial intensity during time-lapse series. The average of 15 distinct cells is shown. Error-bars indicate standard error mean (SEM).

DOI: https://doi.org/10.7554/eLife.31058.017

(*Figure 8A–B*) (*Ringgaard et al., 2011*). Although polar foci were also often observed in a *parP2PM* background, a significantly higher proportion (~20%) of cells exhibited diffuse localization of YFP-ParC compared to wild-type (~4%) (*Figure 8A–B*). Furthermore, in the *parP2PM* strain, a larger proportion of YFP-ParC was diffusely localized in the cytoplasm and there was a significant reduction in the intensity of polar ParC foci (*Figure 8C*). Thus, integration of ParP within signaling arrays via its AIF-domain promotes the retention of ParC at the cell pole. Altogether these data reveal that integration of ParP within signaling arrays not only couples chemotaxis array formation and localization but also modifies the dynamic localization of factors that govern cell pole development, such as ParC and ParP itself.

## Discussion

The highly ordered structure and distribution of signaling arrays within the cell is essential for proper chemotactic responses and bacterial competitiveness. However, it is not well understood how factors responsible for array positioning are able to access chemotaxis proteins within arrays to mediate localization. In addition to interacting with ParC and CheA, we found that ParP also interacts with

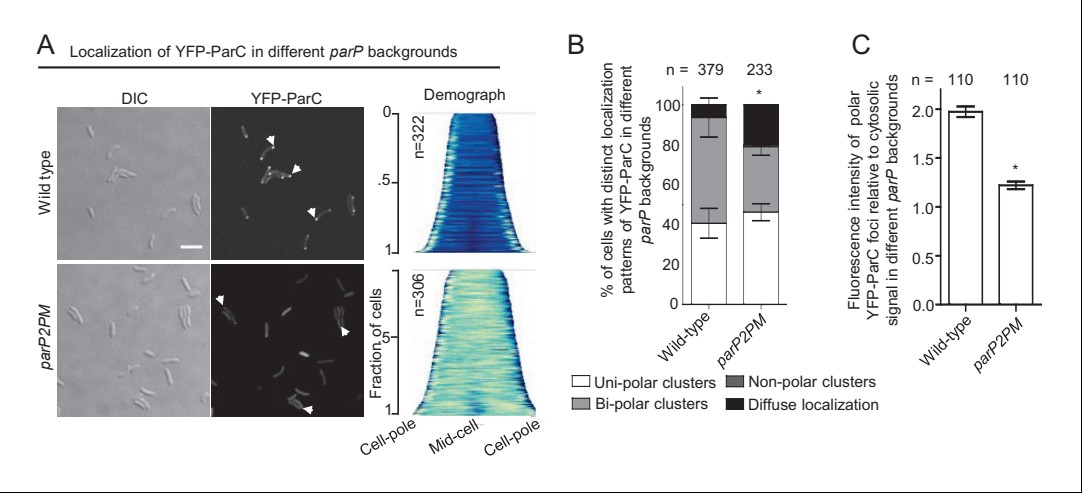

**Figure 8.** Integration of ParP within signaling arrays is required for proper polar localization of ParC. (**A**) Fluorescence microscopy images showing the intracellular localization of YFP-ParC in wild-type and *parP2PM V. cholerae*. Demographs show the fluorescence intensity of YFP along the cell length in a population of *V. cholerae* cells relative to cell length. Scale bar represents 5 μm. White arrows indicate polarly localized ParC clusters. (**B**) Bar graph showing the percentage of cells with distinct YFP-ParC localization patterns in the indicated *V. cholerae* strain backgrounds. (**C**) Bar graph showing the fluorescence intensity of polar ParC foci relative to cytosolic signal. (**B–C**) Error bars indicate standard error of the mean (SEM). The n-value indicates the total number of cells analyzed from three independent experiments. Asterisks indicate p<*0.001* compared wild-type.

DOI: https://doi.org/10.7554/eLife.31058.018

MCP proteins. ParP interacts with the MCP interaction tip domain in a manner analogous to CheA and CheW proteins and promotes array formation. Thus, ParP, like CheA and CheW, appears to be a component of the chemotaxis core unit. Mapping of ParP's interaction interfaces revealed that its C-terminal SH3-like AIF domain includes distinct surfaces that enable interaction with MCPs and CheA. Furthermore, its N-terminal ParC interaction domain is responsible for recruitment of ParP and signaling arrays to the cell pole. The linkage of these domains within ParP couples array formation and localization and results in localized formation of arrays at the cell poles. By stimulating polar array formation, ParP also promotes its own and ParC's polar localization. Collectively, by defining ParP's interaction network and interfaces, we uncovered how this protein couples array formation and polar localization.

We identified MCP proteins as essential interaction partners for ParP. The screen identified 15 distinct MCPs, and all but two MCPs of *V. cholerae* possess the motif within the conserved protein interaction tip that mediates ParP-MCP interaction. The screen likely only identified 15 of the 45 predicted MCPs encoded by *V. cholerae* because only ~50,000 colonies were screened, and thus the screen was not comprehensive. Furthermore, the screen only identifies ParP interaction partners that were fused in frame with the *t18* gene during library generation. These factors likely explain why a subset of MCPs, as well as known interaction partners ParC and CheA were not identified in the screen.

ParP integrates into signaling arrays through its interactions via its AIF domain with the conserved protein interaction tip of MCP proteins and with CheA. Through these complex interactions, ParP promotes array formation rather than compromising array structure. ParP-AIF's similarity to CheW and CheA-P5 in regions that mediate interaction with MCP suggests that ParP might compete with CheW and CheA-P5 for MCP binding, and thereby to become part of the chemotactic core unit. Previous studies indicate that other proteins with SH3-like structures compete to become part of the array in a comparable manner (*Levit et al., 2002*; *Asinas and Weis, 2006*; *Erbse and Falke, 2009*); e.g., CheV can replace CheW (*Alexander et al., 2010*) and CheA-P5 (*Briegel et al., 2012*) within signaling arrays. Although CheW, CheA-P5 and ParP-AIF all appear to have the capacity to recognize and bind the MCP interaction tip within the chemotactic core unit, all have distinct functions and interestingly form their own distinct clades of SH3-like domains. Thus, evolution has exploited the ability of the SH3 domain to interact with MCPs within the core unit of arrays for diverse functions including signal transduction, array formation and the intracellular localization of signaling arrays.

Transactions between ParP and its array partners - MCPs and the LID domain of CheA - likely reflect the balancing of the requirement for an additional array component mediating array localization (ParP) with preservation of array structure and function. Arrays can still form if one of ParP or CheA is absent, due to the presence of the other. owever, in the absence of both proteins, the chemotaxis core unit is no longer assembled and signaling arrays are barely able to form. Arrays form at almost wild-type levels in the absence of CheA, suggesting that ParP is able to fully replace CheA within the core unit. However, we cannot rule out the possibility that ParP also is able to compete with CheW for integration into the core unit. In the prevailing model of array structure, two CheA proteins are present within a core unit, dimerized through their P3 domain (*Figure 9*, type #1) – an interaction that contributes to array stability and signal transduction (*Briegel et al., 2011*; *Li and Hazelbauer, 2011*; *Briegel et al., 2012*; *Li et al., 2013*; *Briegel et al., 2014b*). If ParP-AIF replaces a CheA protein within the arrays, CheA dimerization, and its associated array stabilization, would be lost; however, it was shown that the ParP-CheA interaction (via CheA-LID) reduces dissociation of CheA from arrays (*Ringgaard et al., 2014*) and thereby provides an alternate means of stabilization. Thus, our data are consistent with a model where ParP-AIF replaces some CheA-P5 in binding the MCP interaction tip within the chemotaxis core unit, and is tethered there through binding to the LID domain of the remaining CheA of the core unit. Presumably this AIF-LID interaction is able to substitute for the absence of CheA-P3 dimerization within the core unit and thus maintains the stability of the array structure (*Figure 9*, type #2). Since ParP is able to dimerize via its N-terminus (*Ringgaard et al., 2014*), it is also possible that a ParP dimer is able to replace the CheA dimer within the core unit of wild-type cells, resulting in a core unit comprised of two CheWs and a ParP dimer (*Figure 9*, type #3). Core units consisting only of CheW, ParP, and MCPs presumably constitute the arrays observed within a CheA-deficient strain, which form at close to wild-type levels. Additional studies will be required to elucidate whether interactions between ParP/CheW and ParP/

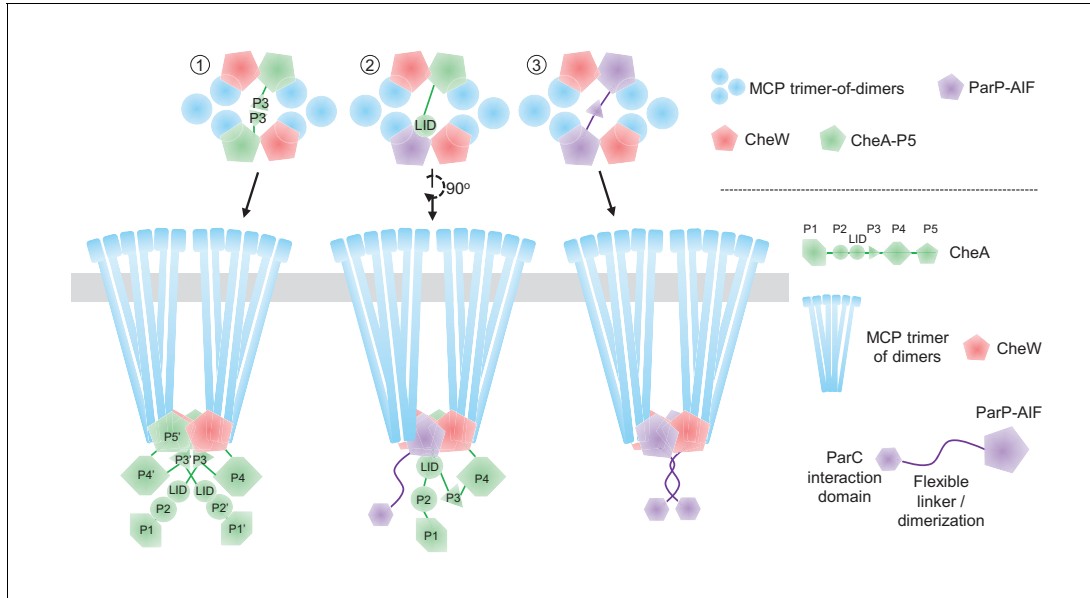

**Figure 9.** Model of ParP's integration within the chemotactic core unit. Schematic model of potential ParP interactions in the chemotaxis core unit – discussed in detail in the main text.

DOI: https://doi.org/10.7554/eLife.31058.019

The following figure supplement is available for figure 9:

**Figure supplement 1.** Chemotaxis core-unit- and array structure in different strain backgrounds.

DOI: https://doi.org/10.7554/eLife.31058.020

CheA-P5 occur and the factors that modulate the MCP interaction tip's accessibility to different partners. In a wild-type background arrays might consist of all three types of core units (*Figure 9—figure supplement 1*), whereas in the ΔparP and ΔcheA deletion backgrounds, arrays consist of type #1 and type #3 core units only, respectively (*Figure 9—figure supplement 1*). Furthermore, it is also a possibility that the flexible dimerization domain of ParP could link ParP proteins from neighboring core units, and in this way augment retention of ParP itself and chemotaxis proteins within the array, thereby contributing to array stability and ultimately their sequestration at the cell pole via ParP's ParC interaction domain.

Interestingly, the intracellular localization of cytosolic chemotaxis arrays in *R. sphaeroides* may have functional similarities to the *Vibrio* ParP/ParC system. In *R. sphaeroides*, a ParA-like protein, PpfA, (ParC is also a ParA-like protein) is thought to ensure proper segregation and positioning of the cytosolic chemotaxis arrays over the bacterial nucleoid by means similar to that used by ParA proteins involved in plasmid segregation (*Thompson et al., 2006*; *Ringgaard et al., 2011*; *Roberts et al., 2012*). PpfA mediates array localization in concert with a predicted cytoplasmic chemoreceptor TlpT, which, in combination with a CheW, is required for array formation. TlpT interacts with PpfA via its N-terminus, thereby likely stimulating PpfA ATPase activity – an action that is required for PpfA function (*Wadhams et al., 2005*; *Thompson et al., 2006*; *Roberts et al., 2012*). It is possible that TlpT integrates into the cytoplasmic arrays, bridging chemotaxis proteins and PpfA to promote array formation and localization in a manner similar to that of ParP.

Importantly, ParP's interactions with the chemotaxis proteins in the core unit (CheA and MCPs) are also important for ParC-mediated sequestration of ParP at the poles and for the polar localization of ParC itself. Disruption of ParP's interactions with MCPs and CheA resulted in a much higher percentage of non-polar (cytosolic) ParC and ParP. Thus, although ParC is still able to recruit ParP to the cell poles, sequestration of ParC and ParP at this site is diminished when ParP interactions with MCPs and CheA are disrupted. As seen with ParC (*Ringgaard et al., 2011*), here we show that there is a continuous exchange of ParP between the cell pole and the cytoplasm. Our photobleaching-based comparisons of ParP and ParP2PM suggest that ParP's capacity to integrate into signaling arrays does not influence its recruitment to the cell pole. In contrast, the capacity of ParP to bind

MCPs and CheA and integrate into signaling arrays had a significant impact on its release from the pole into the cytoplasm. Particularly, that integration of ParP into signaling arrays prevents the release of ParP molecules from the cell pole and consequently promotes its retention at this site. Thus, ParP's integration into arrays modifies its own and likely in turn ParC's subcellular localization dynamics, promoting their polar retention.

We have shown that ParP is a protein of high network connectivity and functions as an important nexus that facilitates chemotaxis array formation, array localization, and regulates the localization dynamics of its network elements. ParP retains partial function as long as one of its network connections to the core unit exists (i.e. to either CheA or MCPs). Only loss of both ParP's connections to the core unit results in a non-functional ParP variant. In contrast, when either of its connections to the core unit are disrupted, the retention of ParP, and ParC, at the cell pole is compromised to the same extend as when both connections simultaneously are disrupted. This suggests that when a ParP loses a network connection to the core unit, ParP is still able to function partially in mediating polar array localization due to the other connection, however, the ability of ParP to mediate retention of its network constituents at the cell pole is lost – ultimately resulting in its partial loss of function. This further emphasizes the importance of ParP's interconnectivity within the chemotaxis protein interaction network in regulating the polar retention of itself and its network constituents and in mediating the proper polar localization of chemotactic signaling arrays.

Taken together our findings show that ParP's high connectivity allows it to serve as a critical nexus that regulates the temporal dynamics of its network constituents and stabilizes the polar localization of the cell-pole anchor ParC and itself. Furthermore, it facilitates the localized assembly and inheritance of signaling arrays at the pole, hereby ensuring proper cell pole development.

# Materials and methods

## Key resources table

| Reagent type or resource | Designation | Source or reference | Identifiers | Additional information |
|---|---|---|---|---|
| Gene (*Vibrio cholerae*) | *parP* | N/A | NCBI-GeneID: 2613440 | |
| Gene (*V.cholerae*) | *cheA* | N/A | NCBI-GeneID: 2613443 | |
| Gene (*V. cholerae*) | *vca0068* | N/A | NCBI-GeneID: 2612100 | |
| Gene (*V. cholerae*) | *vc1868* | N/A | NCBI-GeneID: 2613622 | |
| Gene (*V.cholerae*) | *cheW* | N/A | NCBI-GeneID: 2613439 | |
| Gene (*V. cholerae*) | *parC* | N/A | NCBI-GeneID: 2613441 | |
| Gene (*V. cholerae*) | *mcp; vc1898* | N/A | NCBI-GeneID: 2613527 | |
| Gene (*V. cholerae*) | *vca0658* | N/A | NCBI-GeneID: 2612769 | |
| Strain (*V. cholerae* N16961) | wild type | Clinical isolate | NCBI-Taxonomy ID: 243277 | For complete strain list see *Supplementary file 1* |
| Antibody | Mouse monoclonal anti-GFP | Clontech Laboratories, Inc. (USA) | Cat#: 632381 | |
| Recombinant DNA reagent | | | | See *Supplementary file 2* for Plasmid list |
| Sequence-based reagents | Oligonucleotides | This paper | See *Supplementary file 1* for Primer list | Eurofins MWG Operon (Ebersberg) |
| NucleoSpin Gel and PCR Clean-up kit | N/A | Macherey-Nagel (Düren) | Ref.: 740609.250 | |
| NucleoSpin Plasmid Kit | N/A | Macherey-Nagel (Düren) | Ref.: 740588.250 | |
| Metamorph v7.5 | N/A | Molecular Devices (Union City, CA) | | |
| ImageJ-Fiji | N/A | http://rsbweb.nih.gov/ij | | |
| R studio version 3.0.1 | N/A | http://www.rstudio.com/ | | |

*Continued on next page*

*Continued*

| Reagent type or resource | Designation | Source or reference | Identifiers | Additional information |
|---|---|---|---|---|
| GraphPad Prism version 6.07 | N/A | GraphPad Software (La Jolla CA) | | |
| NIS-Elements Software AR 4.60.00 (Nikon) | N/A | LIM (Prague) | ' | |

## Growth conditions and media

If not otherwise stated, *V. cholerae* and *E. coli* were grown in LB media or on LB agar plates at 30°C or 37°C containing antibiotics in the following concentrations: streptomycin 200 µg/ml; kanamycin 50 µg/ml; ampicillin 100 µg/ml; chloramphenicol 20 µg/ml for *E. coli* and 5 µg/ml for *V. cholerae*. When needed, L-arabinose was added to a final concentration of 0.2 % w/v.

## Strains

*E. coli* strain DH5α*λpir* was used for cloning and *E. coli* strain SM10*λpir* was used to transfer plasmid DNA by conjugation from *E. coli* to *V. cholerae* (**Miller and Mekalanos, 1988**). The wild-type strain of *V. cholerae* used was the El Tor clinical isolate N16961 and all mutants are derivatives of this strain. Construction of *V. cholerae* deletion or point mutants was performed with standard allele exchange techniques using derivatives of plasmid pCVD442 (**Donnenberg and Kaper, 1991**). All strains used are listed in **Supplementary file 1**.

## Plasmids

All plasmids used in this study are listed in **Supplementary file 1**. All primers used in construction of plasmids are listed in **Supplementary file 2**.

### Plasmids pAK2 and pAK8

The gene for *vc2060* was PCR amplified from *V. cholerae* using primers VC2060-BTH-cw/VC2060-BTH-ccw. The PCR product was digested with BamHI and EcoRI and was inserted into the equivalent sites of plasmids pUT18C and pKT25 resulting in plasmids pAK2 and pAK8 respectively.

### Plasmid pAK7

The gene for *vc2059* was PCR amplified from *V. cholerae* using primers VC2059-BTH-cw/VC2059-BTH-ccw. The PCR product was digested with BamHI and KpnI and was inserted into the equivalent sites of plasmid pKT25 resulting in plasmid pAK7.

### Plasmid pAK9

The gene for *vc2061* was PCR amplified from *V. cholerae* using primers VC2061-BTH-cw/VC2061-BTH-ccw. The PCR product was digested with BamHI and KpnI and was inserted into the equivalent sites of plasmid pKT25 resulting in plasmid pAK9.

### Plasmid pAK10

The gene for *vc2063* was PCR amplified from *V. cholerae* using primers VC2063-BTH-cw/VC2063-BTH-ccw. The PCR product was digested with XbaI and KpnI and was inserted into the equivalent sites of plasmid pKT25 resulting in plasmid pAK10.

### Plasmid pAK14

The gene for *vc2063* was PCR amplified from *V. cholerae* using primers VC2063-1-cw/VC2063-1-ccw. The PCR product was digested with XbaI and SphI and was inserted into the equivalent sites of plasmid pMF390 resulting in plasmid pAK14.

### Plasmid pAK63

The gene coding for amino acids 1–628 of CheA1, which constitutes domains P1 to P4 (*vc2063*, base pairs 1–1884) was PCR amplified from *V. cholerae* using primers VC2063-1-cw/VC2063-7-ccw. The

PCR product was digested with XbaI and SphI and was inserted into the equivalent sites of plasmid pMF390 resulting in plasmid pAK63.

## Plasmid pAK72

The gene coding for amino acids 643–785 of CheA1, which constitutes domain P% (*vc2063*, base pairs 1929–2355) was PCR amplified from *V. cholerae* using primers VC2063-8-cw/VC2063-1-ccw. The PCR product was digested with XbaI and SphI and was inserted into the equivalent sites of plasmid pMF390 resulting in plasmid pAK72.

## Plasmid pAK13

The up- and downstream regions flanking *vc2063* were amplified using primer pairs vc2063-del-a/vc2063-del-b and vc2063-del-c/vc2063-del-d, respectively, using *V. cholerae* chromosomal DNA as template. In a third PCR, using primers vc2063-del-a/vc2063-del-d and products of the first two PCR reactions as template, the flanking regions were stitched together. The resulting product was digested with XbaI and was inserted into the equivalent site of pCVD442, resulting in plasmid pAK13.

## Plasmids pAK80 and pAK90

The gene coding for amino acids 461–672 of MCP VC1898 (*vc1898*, base pairs 1383–2016) was PCR amplified from *V. cholerae* using primers VC1898-cw2/VC1898-ccw1. The PCR product was digested with XbaI and KpnI and was inserted into the equivalent sites of plasmids pKT25 and pUT18C resulting in plasmids pAK80 and pAK90 respectively.

## Plasmid pAK84

The gene coding for amino acids 330–547 of MCP VCA0068 (*vca0068*, base pairs 990–1641) was PCR amplified from *V. cholerae* using primers VCA0068-cw2/VCA0068-ccw1. The PCR product was digested with BamHI and KpnI and was inserted into the equivalent sites of plasmid pUT18C resulting in plasmid pAK84.

## Plasmid pAK86

The gene coding for amino acids 335–536 of MCP VA0658 (*vca0658*, base pairs 1005–1608) was PCR amplified from *V. cholerae* using primers VCA0658-cw2/VCA0658-ccw1. The PCR product was digested with BamHI and KpnI and was inserted into the equivalent sites of plasmid pUT18C resulting in plasmid pAK86.

## Plasmid pAK88

The gene coding for amino acids 424–626 of MCP VC1868 (*vc1868*, base pairs 1272–1878) was PCR amplified from *V. cholerae* using primers VC1868-cw2/VC1868-ccw1. The PCR product was digested with BamHI and KpnI and was inserted into the equivalent sites of plasmid pUT18C resulting in plasmid pAK88.

## Plasmid pSR1218

Amino acid substitution L196A was introduced in ParP using plasmid pAK2 as template and rolling circle PCR using primers vc2060-L196A-cw/vc2060-L196A-ccw, resulting in plasmid pSR1218.

## Plasmid pSR1219

Amino acid substitution L209A was introduced in ParP using plasmid pAK2 as template and rolling circle PCR using primers vc2060-L209A-cw/vc2060-L209A-ccw, resulting in plasmid pSR1219.

## Plasmid pSR1220

Amino acid substitution L212A was introduced in ParP using plasmid pAK2 as template and rolling circle PCR using primers vc2060-L212A-cw/vc2060-L212A-ccw, resulting in plasmid pSR1220.

### Plasmid pSR1221
Amino acid substitution I215A was introduced in ParP using plasmid pAK2 as template and rolling circle PCR using primers vc2060-I215A-cw/vc2060-I215A-ccw, resulting in plasmid pSR1221.

### Plasmid pAA44
Plasmid pAA44 was constructed by PCR amplification of the up- and down-stream regions of *vc2060* encoding the AIF domain using *V. cholerae* chromosomal DNA as template. In a third PCR reaction the part of *vc2063* encoding the P5 domain was amplified using *V. cholerae* chromosomal DNA as template. PCR1 and PCR2 were performed with primer pairs VC2060_ CheWlike-XbaI-a1/VC2060-VC2063-b1 and VC2060-VC2063-e1/VC2060_CheWlike-XbaI–f1 respectively. PCR3 was performed with primer pair VC2063_P5-c1/VC2063_P5-d1. A fourth PCR was then performed using primer pair VC2060_ CheWlike-XbaI-a1/VC2060_CheWlike-XbaI–f1 and the products of PCR1, PCR2, and PCR3 as template. The resulting PCR product was digested with XbaI and ligated into the equivalent site in pCVD442 resulting in plasmid pAA44.

### Plasmid pAA48
Amino acid substitution L521R was introduced in VC1898 using plasmid pAK90 as template and rolling circle PCR using primers VC1898-L521R-cw/VC1898-L521R-ccw, resulting in plasmid pAA48.

### Plasmid pAA50
Amino acid substitution N522R was introduced in VC1898 using plasmid pAK90 as template and rolling circle PCR using primers VC1898-N522R-cw/VC1898-N522R-ccw, resulting in plasmid pAA50.

### Plasmid pAA51
Amino acid substitution A524R was introduced in VC1898 using plasmid pAK90 as template and rolling circle PCR using primers VC1898-A524R-cw/VC1898-A524R-ccw, resulting in plasmid pAA51.

### Plasmid pAA56
Amino acid substitution L518R was introduced in VC1898 using plasmid pAK90 as template and rolling circle PCR using primers VC1898-L518R-cw/VC1898-L518R-ccw, resulting in plasmid pAA56.

### Plasmid pAA60
The gene encoding for *vc1898* was amplified using primers vc1898-cw and vc1898-ccw using genomic DNA from *Vibrio cholerae* N16961. The resulting PCR fragment was digested with enzymes BsrGI and SphI. Subsequently, the digested fragment was inserted into the equivalent sites of plasmids pJH37 resulting in plasmid pAA60.

### Plasmid pAA74
The genes encoding for *mCherry* and *vc1898* were amplified using primers vc1998-cherry-2-cw and vc1898-XmaI-ccw using plasmid pAA60 as template. The resulting PCR fragment was digested with SphI and XmaI and then inserted into the equivalent sites in plasmid pUC19 (*Norrander et al., 1983*), resulting in plasmid pAA74.

### Plasmid pAA75
The gene encoding for *cheW1* was PCR amplified from plasmid pSR1033 using primers CFP-VC2059-cw and CFP-VC2059-ccw, the resulting fragment was then digested with BsrGI and SphI. Then the digested fragment was inserted in the corresponding sites of plasmid pMF391 (*Yamaichi et al., 2007*), finally resulting in plasmid pAA75.

### Plasmid pAA76, pAA77, pAA78, and pAA79
The gene encoding for *cfp-cheW1* was amplified using primers ShDo-Spc-CFP-CheW and CFP-VC2059-ccw from plasmid pAA75. The resulting fragment was digested with HincII and SphI and inserted into the corresponding sites of plasmids pPM15, pSR1102, pPM14 and pAK105, resulting in plasmids pAA76, pAA77, pAA78 and pAA79, respectively.

### Plasmid pPM010
Amino acid substitution W305A was introduced in ParP using plasmid pAK2 as template and rolling circle PCR using primers vc2060-W305A-cw/vc2060-W305A-ccw, resulting in plasmid pPM010.

### Plasmid pPM011
Amino acid substitution W305A was introduced in ParP using plasmid pSR1219 as template and rolling circle PCR using primers vc2060-W305A-cw/vc2060-W305A-ccw, resulting in plasmid pPM011.

### Plasmid pPM014
The gene for *vc2060W305A* was PCR amplified from plasmid pPM010, using primers VC2060-1-cw/VC2060-1-ccw. The PCR product was digested with BsrG1 and HincII and was inserted into the equivalent sites of plasmid pMF390, resulting in plasmid pPM014.

### Plasmid pPM015
The gene for *vc2060W305A* was PCR amplified from plasmid pPM011, using primers VC2060-1-cw/VC2060-1-ccw. The PCR product was digested with BsrG1 and HincII and was inserted into the equivalent sites of plasmid pMF390, resulting in plasmid pPM014.

### Plasmid pPM020
Plasmid pPM020 was constructed by PCR amplification of the up- and down-stream regions of vc2060 using *V. cholerae* chromosomal DNA as template. In a third PCR reaction *vc2060W305A* was amplified using plasmid pPM010 as template. PCR1 and PCR2 were performed with primer pairs VC2060-PM-ins-a/VC2060-PM-ins-b and VC2060-PM-ins-e/VC2060-PM-ins-f respectively. PCR3 was performed with primer pair VC2060-PM-ins-c/VC2060-PM-ins-d. A fourth PCR was then performed using primer pair VC2060-PM-ins-a/VC2060-PM-ins-f and the products of PCR1, PCR2, and PCR3 as template. The resulting PCR product was digested with SacI and ligated into the equivalent site in pCVD442 resulting in plasmid pPM020.

### Plasmid pPM021
Plasmid pPM021 was constructed by PCR amplification of the up- and down-stream regions of vc2060 using *V. cholerae* chromosomal DNA as template. In a third PCR reaction *vc2060L209A* was amplified using plasmid pSR1219 as template. PCR1 and PCR2 were performed with primer pairs VC2060-PM-ins-a/VC2060-PM-ins-b and VC2060-PM-ins-e/VC2060-PM-ins-f respectively. PCR3 was performed with primer pair VC2060-PM-ins-c/VC2060-PM-ins-d. A fourth PCR was then performed using primer pair VC2060-PM-ins-a/VC2060-PM-ins-f and the products of PCR1, PCR2, and PCR3 as template. The resulting PCR product was digested with SacI and ligated into the equivalent site in pCVD442 resulting in plasmid pPM021.

### Plasmid pPM027
Plasmid pPM027 was constructed by PCR amplification of the up- and down-stream regions of vc2060 using *V. cholerae* chromosomal DNA as template. In a third PCR reaction *vc2060L209A-W305A* was amplified using plasmid pPM011 as template. PCR1 and PCR2 were performed with primer pairs VC2060-PM-ins-a/VC2060-PM-ins-b and VC2060-PM-ins-e/VC2060-PM-ins-f respectively. PCR3 was performed with primer pair VC2060-PM-ins-c/VC2060-PM-ins-d. A fourth PCR was then performed using primer pair VC2060-PM-ins-a/VC2060-PM-ins-f and the products of PCR1, PCR2, and PCR3 as template. The resulting PCR product was digested with SacI and ligated into the equivalent site in pCVD442 resulting in plasmid pPM027.

## Bacterial-two-hybrid screen
A schema explaining the bacterial-two-hybrid screen is presented in *Figure 2A* and *Figure 2—figure supplement 1*. *V. cholerae* chromosomal DNA was digested with rare-cutter restriction enzymes and fragments in the size-range 1000–5000 bp purified and fused to the gene encoding the T25 fragment of adenylate-cyclase in vector pKT25, thereby resulting in a library of chromosomal DNA fused to the gene encoding T25. The library was transformed into *E. coli* strain BTH101

(*Karimova et al., 1998*) expressing T18-ParP (plasmid pAK2) and transformants were spread on indicator plates. One hundred blue colonies were screened by sequencing for identification of the chromosomal DNA insert in pKT25 encoding a possible ParP interaction partner.

## Fluorescence microscopy

Fluorescence microscopy was carried out essentially as described in references (*Ringgaard et al., 2015*; *Briegel et al., 2016*). For fluorescence microscopy in *V. cholerae*, fluorescent fusion proteins were ectopically expressed from plasmids. Cells were grown for 12 hr in LB medium at 37° C with shaking. Ten microliters were then used to inoculate 5 milliliter cultures. When OD600 ≈ 1.0, protein expression was induced by addition of 0.2% w/v final concentration of L-arabinose. The cultures were incubated for one additional hour, at which point cells were ready for microcopy analysis.

For fluorescence microscopy of *E. coli* strain VS296, a strain carrying the relevant plasmid for fluorescent protein expression was inoculated in 5 mL 10% LB in PBS buffer. Expression of fluorescence proteins was induced by addition of 0.4% w/v final concentration of L-arabinose and 1 mM IPTG. Cultures were incubated 8–10 hr, at which time-point cells were ready for microscopy analysis.

Cells ready for microscopy analysis were mounted onto 1% agarose (in 20% PBS buffer with 10% LB) on a microscopy slide before imagining. Microscopy of YFP-CheW1 was performed using a Zeiss Axio Imager M1 fluorescence microscope. Images were collected with a Cascade:1K CCD camera (Photometrics), using a Zeiss αPlan-Fluar 100x/1.45 Oil DIC objective. Images were analyzed using MetaMorph (version 7.7.5.0; Molecular Devices). Imaging of YFP-CheA1 variants was performed using a Zeiss Axioplan 2 microscope equipped with a 100 × a plan lens andHamamatsu cooled CCD camera. All other microscopy was performed using a Nikon eclipse Ti inverted Andor spinning-disc confocal microscope equipped with a 100x lens and an Andor Zyla sCMOS cooled camera and an Andor FRAPPA system. Microscopy images were analyzed using ImageJ imaging software (http://rsbweb.nih.gov/ij) and Metamorph Offline (version 7.7.5.0, Molecular Devices). For comparison, all mutant strains were imaged with the same exposure time and light intensity as the wild-type background. Generation of demographs was carried out precisely as described by (*Cameron et al., 2014*; *Heering and Ringgaard, 2016*; *Heering et al., 2017*).

## Sample size and analysis

For microscopy experiments counting the percentage of cells with distinct localization patterns a minimum of three biological experiments were performed and for each experiment >100 cells were counted in order to determine the percentages of cells with different localization patterns. Cells and localization patterns were enumerated by hand. The mean of the three experiments was then plotted with error bars indicating the standard-error-mean (SEM). A t-test was performed to calculate the *p* value. The n-value indicates the total number of cells analyzed of the three independent experiments and is included for each sample in the respectively figures.

For microscopy experiments measuring the fluorescence intensity of polar foci relative to the cytosolic signal (*Figures 7C* and *8C*), relative intensity was measured in the total number of cells indicated (n) in the respective figures. The mean was then plotted with error bars representing the SEM. The *p*-value was calculated performing a Student's t-test.

For demographic analysis the data from three biological experiments were pooled and for each experiment >100 cells were analyzed. The total number of cells included (n) is mentioned for each demograph in the respective figures.

## Western-blot analysis

To test for stability and expression of YFP-CheW1, bacterial samples were collected from cultures ready for fluorescence microscopy analysis. Samples from different strains were normalized to optical density and subjected to western-blot analysis using JL8 anti-GFP antibodies (also recognizing YFP). As a positive control, a strain only expressing YFP was included. Additionally, a strain not expressing any YFP variant was included as a negative control.

## Photobleaching time-lapse experiments

Photobleaching time-lapse experiments were performed using the Andor FRAPPA system. Cells were treated and mounted on agarose pads as described for fluorescence microscopy of *V. cholerae*

cells. For FRAP experiments, a point-of-interest was bleached using a 515 nm laser at 7% intensity. For bleaching of the cytoplasm, a region-of-interest corresponding to 2/3 of the cell length was chosen and bleached with 1 pulse using a 515 nm laser at 7% intensity. Cells were then imaged over time. For each time-point the fluorescence intensity at the cell pole was then calculated relative to the pre-bleach intensity and plotted as a function of time. Graphs represent the average intensity of the indicated number of cells analyzed, with error-bars representing SEM. The total number of cells analyzed (n) is mentioned in the figure.

## Multiple sequence alignment and generation of phylogenetic trees

In generation of multiple sequence alignments of *V. cholerae* MCPs and ParP orthologues, respectively, we used the MUSCLE tool at default settings (*Edgar, 2004*). Phylogenetic trees were generated based on MUSCLE sequence alignments using Jalview Average Distance BLOSOM62 with default settings. Phylogenetic trees generated in Jalview were displayed and colored using iTOL (*Letunic and Bork, 2011*).

ParP orthologs, in generation of the sequence alignment in *Figure 3—figure supplement 2* and phylogenetic tree in *Figure 3A*, were chosen based on a STRING (*Jensen et al., 2009*) analysis of ParP and ParC from *V. cholerae*, using default settings. Thus, all ParP orthologs included in the analysis are encoded by predicted *parP* genes, located within a chemotaxis operon, and with an associated *parC* gene immediately upstream, indicating that all ParPs included in the analysis are part of a ParC/ParP-system. Furthermore, in generation of the phylogenetic tree in *Figure 3A*, we included CheWs and CheAs from chemotaxis operons that have an associated ParC/ParP-system, based on a STRING analysis of ParP and ParC from *V. cholerae*.

In *Figure 3—figure supplement 1*, ParP from *V. parahaemolyticus* was aligned against CheW from *T. maritima* MSB8. *T. maritima* MSB8 CheW was chosen as reference for the alignment as amino acid residues from *T. maritima* MSB8 CheW important for mediating interactions to MCPs have been solved (*Griswold et al., 2002*; *Park et al., 2006*; *Briegel et al., 2012*; *Li et al., 2013*).

## Cryo electron microscopy

For imaging, *V. cholerae* strains were cultured overnight in 5 ml LB media at 37°C with 200 rpm shaking. For each strain, 3 µl cell culture were applied to a freshly plasma-cleaned R2/2 copper Quantifoil grid (Quantifoil Micro Tools, Jena, Germany). Plunge freezing was carried out with a Leica EMGP (Leica microsystems, Wetzlar, Germany). Excessive liquid was wicked off from the grid by 1 s blotting inside the chamber set at room temperature and 95% humidity. Grids were plunge frozen in liquid ethane at −183°C and then stored in liquid nitrogen until imaging. Cryo EM images were collected on a Talos L120C transmission electron microscope (Thermo Fisher Scientific (formerly FEI), Hillsboro, OR, USA) operating at 120 kV. All targets were randomly picked, manually located and imaged in low dose mode.

## Acknowledgements

We are grateful to Kathrin Schirner for comments on the manuscript and suggestions for experiments. We thank Brigid Davis for comments on the manuscript. We thank Victor Sourjik for donation of *E. coli* strain VS296. This work was supported by the Max Planck Society and grant RI 2820/1–1 from the DFG, Deutsche Forschungsgemeinschaft (SR) and NIH R37 AI-042347 and HHMI (MKW).

## Additional information

### Funding

| Funder | Grant reference number | Author |
| --- | --- | --- |
| Max-Planck-Institut für Terrestrische Mikrobiologie | Open-access funding | Simon Ringgaard |
| Deutsche Forschungsgemeinschaft | RI 2820/1-1 | Simon Ringgaard |
| National Institutes of Health | NIH R37 AI-042347 | Matthew K Waldor |

| Howard Hughes Medical Institute | Matthew K Waldor |

The funders had no role in study design, data collection and interpretation, or the decision to submit the work for publication.

## Author contributions
Alejandra Alvarado, Formal analysis, Investigation, Writing—review and editing, Data interpretation and generation of working model; Andreas Kjær, Wen Yang, Formal analysis, Investigation; Petra Mann, Investigation; Ariane Briegel, Formal analysis, Investigation, Writing—review and editing; Matthew K Waldor, Conceptualization, Funding acquisition, Writing—review and editing; Simon Ringgaard, Conceptualization, Formal analysis, Supervision, Funding acquisition, Investigation, Methodology, Writing—original draft, Project administration, Writing—review and editing, Data interpretation and generation of working model

## Author ORCIDs
Matthew K Waldor (iD) https://orcid.org/0000-0003-1843-7000
Simon Ringgaard (iD) http://orcid.org/0000-0002-4980-5964

## Decision letter and Author response
Decision letter https://doi.org/10.7554/eLife.31058.025
Author response https://doi.org/10.7554/eLife.31058.026

## Additional files

### Supplementary files
• Supplementary file 1. Table S1 - strain and plasmid list. Table listing the strains and plasmids used in the study.
DOI: https://doi.org/10.7554/eLife.31058.021

• Supplementary file 2. Table S2 - primer list. Table listing the primer names and sequences used in the study.
DOI: https://doi.org/10.7554/eLife.31058.022

• Transparent reporting form
DOI: https://doi.org/10.7554/eLife.31058.023

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
