## [Decision Letter]

Thank you for submitting your article "Coupling chemosensory array formation and localization" for consideration by *eLife*. Your article has been favorably evaluated by Gisela Storz (Senior Editor) and three reviewers, one of whom is a member of our Board of Reviewing Editors. The following individual involved in review of your submission has agreed to reveal his identity: Grant Bowman (Reviewer #2).

The reviewers have discussed the reviews with one another and the Reviewing Editor has drafted this decision to help you prepare a revised submission.

The authors present a detailed study that addresses some important unanswered questions about the mechanism of chemotaxis array assembly. It makes significant strides in elucidating the role of ParP, a protein currently known as being important for array assembly and localization, yet its specific role is unclear. The manuscript identifies ParP as a central component in the core complex of the chemoreceptor array, by providing strong evidence that this protein participates in physical interactions with multiple core components. Interestingly, the results suggest competition between ParP, CheA, and CheW for binding to MCP tip, and that the same domain of ParP contains two genetically and functionally separable binding sites for MCP and CheA. The resulting model, though still incomplete insofar as describing a complete view of array formation (and especially the polar localization of assembled arrays), makes significant advancements in understanding the complex and interesting problem of polar chemotaxis array assembly.

Some modifications should be however made for the paper to be acceptable for publication:

- The authors refer to "chemosensory arrays" when they only observe fluorescent clusters throughout the study. Granted these clusters might reflect ordered arrays but it is also possible that they represent supramolecular complexes in various states, which might change depending on the genetic background. For example, what is the evidence that the chemotaxis proteins are indeed organized in hexagonal arrays in a CheA mutant? From the study, this conclusion is only inferred from a polar CheW-YFP cluster. The authors should make it clear when an ordered array is hypothesized.

- All reviewers agree that the FRAP analysis only shows modest differences that could have methodological origins. As presented, the FRAP data only shows that Polar ParP (WT and mutant) do exchange quite rapidly with the polar pool. The FRAP analysis could be deleted. Alternatively it should be expanded, perhaps by performing FRAP experiments on CheW-YFP clusters and explore the supramolecular complexes discussed above ?

- Figure 7: In this experiment, the P2M mutant did not show an increased effect over single mutants, but in Figure 4 and Figure 6, the additive effect of the P2M mutant is apparent. What could account for this difference? If the issue is not easily addressed in the manuscript text, perhaps the experiment in Figure 7 could be repeated in a *∆parC ∆parP* double knockout strain. This might make it possible to determine if it is dependent on ParP self-interaction or interaction with ParC.

- Discussion: The difference in the connections among ParP, CheW, and CheA in core chemotaxis units (as shown in Figure 1 and Figure 9) versus their role in promoting interconnected arrays of core units (as shown in Figure 1) is confusing. Is it possible that ParP and / or CheA could be participating in one type of interaction but not the other? Could the loose dimerization domain of the ParP element shown in model 9-2 participate in linking with ParP proteins in neighboring arrays?

- A description of the methods used to make the phylogenetic tree in Figure 3, or the reasons why representatives from the species shown in the multispecies alignments in Figure 2—figure supplement 3 and Figure 3—figure supplement 2 were chosen, is missing. Further, it is important to mention in the manuscript that *E. coli* does not have ParP or ParC homologs, because this greatly reduces the possibility that the interactions identified in the BACTH assay occur indirectly through a hypothetical additional factor, endogenous to *E. coli*, that exists for chemotaxis array assembly.

---

## [Author Response]

[…] Some modifications should be however made for the paper to be acceptable for publication:- The authors refer to "chemosensory arrays" when they only observe fluorescent clusters throughout the study. Granted these clusters might reflect ordered arrays but it is also possible that they represent supramolecular complexes in various states, which might change depending on the genetic background. For example, what is the evidence that the chemotaxis proteins are indeed organized in hexagonal arrays in a CheA mutant? From the study, this conclusion is only inferred from a polar CheW-YFP cluster. The authors should make it clear when an ordered array is hypothesized.

The reviewers raise a very valid point. We have addressed this point by performing cryo electron microscopy to evaluate array formation and structure in wild-type and *cheA* mutant cells. The results of these experiments are now shown as “Figure 1—figure supplement 1”. The following new paragraph describing our findings has been added to the revised manuscript in the first section of the Results (“ParP contributes to signaling array formation”):

“To analyze if arrays are still properly formed in the absence of CheA, we performed cryo-electron microscopy (cryo-EM) on wild-type and *ΔcheA* cells (Figure 1—figure supplement 1). […] Furthermore, these cryo-EM images strongly suggest that the YFP-CheW1 clusters reflect the localization and formation of properly structured arrays in the absence of CheA, although we cannot formally exclude the possibility that YFP-CheW1 clusters may reflect misformed or variant states of supramolecular complexes in some cells.”

Furthermore, a paragraph has been added to the “Materials and methods” section, which describes the methods used for cryo-EM of *V. cholerae* cells. The cryo-electron microscopy was carried out by Wen Yang and Ariane Briegel at the Institute of Biology, Leiden University, The Netherlands, and as such they have been added to the author list of the manuscript.

- All reviewers agree that the FRAP analysis only shows modest differences that could have methodological origins. As presented, the FRAP data only shows that Polar ParP (WT and mutant) do exchange quite rapidly with the polar pool. The FRAP analysis could be deleted. Alternatively it should be expanded, perhaps by performing FRAP experiments on CheW-YFP clusters and explore the supramolecular complexes discussed above ?

We agree with the reviewers that the FRAP analysis of polar clusters only show very modest differences. We have now performed statistical analysis on the recovery curves for wild-type and ParP2PM in Figure 7. This analysis showed that, indeed, the reviewers are astute; there is no significant difference in recovery between wild-type ParP and ParP2PM (p-value added to the figure). We thank the reviewers for bringing this important point to our attention. However, we still think that the FRAP experiment is important to include in the manuscript, in order to show that there is a continuous recruitment of new ParP protein from the cytoplasm to the cell pole. We have now revised the manuscript accordingly, and it is no longer stated that ParP2PM has a faster recovery rate than wild-type ParP. Furthermore, we performed a significance test on the ParP decay experiment in Figure 7 (p-value added to the figure). The decay experiment shows that ParP2PM has a significantly higher release from the cell pole than wild-type ParP. In fact, we think these changes strengthen the manuscript, since the FRAP experiment shows that integration of ParP into arrays does not affect ParP’s recruitment to the cell pole. While the decay assay shows that ParP’s capacity to interact with MCPs and CheA (and thereby integrate into signaling arrays) prevents its release, and as such promotes its retention at the cell pole, consequently stabilizing its localization at this site. Accordingly, we have changed the title of the paragraph to “Integration of ParP within signaling arrays promotes its retention at the cell pole”. We have also changed the Discussion accordingly to reflect these changes. The Discussion now reads: “As seen with ParC (Ringgaard et al., 2011), here we show that there is a continuous exchange of ParP between the cell pole and the cytoplasm. […] Thus, ParP’s integration into arrays modifies its own and likely in turn ParC’s subcellular localization dynamics, promoting their polar retention.”

- Figure 7: In this experiment, the P2M mutant did not show an increased effect over single mutants, but in Figure 4 and Figure 6, the additive effect of the P2M mutant is apparent. What could account for this difference? If the issue is not easily addressed in the manuscript text, perhaps the experiment in Figure 7 could be repeated in a ∆parC ∆parP double knockout strain. This might make it possible to determine if it is dependent on ParP self-interaction or interaction with ParC.

As the reviewers point out, there is a clear additive effect on combining the L209A and W305A aa-substitutions (ParP2PM) on the ability of ParP to associate with signaling arrays (Figure 4) and to recruit chemotaxis arrays to the cell pole (Figure 6). However, there is no additive effect of combining L209A and W305A on ParP’s own localization at the cell pole, which is almost equally compromised for both single (L209A and W305A, respectively) and double (ParP2PM) aa-substitutions in ParP. It is also important to note that both ParP’s association with arrays and ParP’s recruitment of arrays to the cell pole is compromised when its interaction to either CheA or MCP is disrupted (Figure 4 and Figure 6) – and thus that disruption of either of ParP’s interactions to CheA or MCP results in a partial deficiency in ParP function.

We have added the following paragraph to the Discussion (seventh paragraph): “We have shown that ParP is a protein of high network connectivity and functions as an important nexus that facilitates chemotaxis array formation, array localization, and regulates the localization dynamics of its network elements. […] This, further emphasizes the importance of ParP’s interconnectivity within the chemotactic protein interaction network in regulating the polar retention of itself and its network constituents and in mediating the proper polar localization of chemotactic signaling arrays.”

We believe this point had not been emphasized properly in the manuscript text, and we thank the reviewers for bringing it to our attention.

- Discussion: The difference in the connections among ParP, CheW, and CheA in core chemotaxis units (as shown in Figure 1 and Figure 9) versus their role in promoting interconnected arrays of core units (as shown in Figure 1) is confusing. Is it possible that ParP and / or CheA could be participating in one type of interaction but not the other? Could the loose dimerization domain of the ParP element shown in model 9-2 participate in linking with ParP proteins in neighboring arrays?

In order to clarify the role of ParP and CheA in promoting interconnected arrays of core units, we have added a new Figure (Figure 9—figure supplement 1), which depicts the structure of signaling arrays in wild-type, *ΔparP, ΔcheA*, and *ΔparPΔcheA* strain backgrounds. Furthermore, we agree with the reviewers that it is possible that the loose dimerization domain of the ParP element shown in model 9-2 could participate in linking with ParP proteins in neighboring core units. This is a very important point and we have added the following statement to the Discussion (fourth paragraph): “In a wild-type background arrays might consist of all three types of core units (Figure 9—figure supplement 1), whereas in the *ΔparP* and *ΔcheA* deletion backgrounds, arrays consist of type #1 and type #3 core units only, respectively (Figure 9—figure supplement 1). Furthermore, it is also a possibility that the flexible dimerization domain of ParP could link ParP proteins from neighboring core units, and in this way augment retention of ParP itself and chemotaxis proteins within the array, thereby contributing to array stability and ultimately their sequestration at the cell pole via ParP’s ParC interaction domain.”

- A description of the methods used to make the phylogenetic tree in Figure 3, or the reasons why representatives from the species shown in the multispecies alignments in Figure 2—figure supplement 3 and Figure 3—figure supplement 2 were chosen, is missing. Further, it is important to mention in the manuscript that E. coli does not have ParP or ParC homologs, because this greatly reduces the possibility that the interactions identified in the BACTH assay occur indirectly through a hypothetical additional factor, endogenous to E. coli, that exists for chemotaxis array assembly.

We have now added a paragraph titled “Multiple sequence alignment and generation of phylogenetic trees” to the Materials and methods section of the manuscript. Here we describe the methods used to generate the multiple sequence alignments and phylogenetic trees presented in the study. Furthermore, we describe the reasons why the representatives from the different species were chosen.

In order to make it more clear what MCPs were chosen for the sequence alignment in Figure 2—figure supplement 3 we added the following sentence: “Multiple sequence alignment of all predicted MCPs from *V. cholerae* against the sequence of MCP TM1143 from *T. maritima*, showed that these four residues are conserved in all of the MCPs identified in the two-hybrid screen and all but two putative MCPs found in *V. cholerae* (Figure 2—figure supplement 3)”.

Furthermore, we agree with the reviewers that is important to mention that *E. coli* does not encode any ParP/ParC homologs. Thus, we have added the following statement to the manuscript: “It is important to note that *E. coli* does not encode homologs of either ParP or ParC, thus reducing the possibility of indirect interactions mediated by an endogenous *E. coli*, factor, and suggesting that interaction partners identified in this assay likely interact directly with ParP.”